# TSkips: Efficiency Through Explicit Temporal Delay Connections in Spiking Neural Networks

**Prajna G. Malettira**  *pmaletti@purdue.edu*
*Department of Electrical and Computer Engineering*
*Purdue University*

**Shubham Negi**  *snegi@purdue.edu*
*Department of Electrical and Computer Engineering*
*Purdue University*

**Wachirawit Ponghiran**  *wponghir@ibm.com*
*Department of Electrical and Computer Engineering*
*Purdue University*

**Kaushik Roy**  *kaushik@purdue.edu*
*Department of Electrical and Computer Engineering*
*Purdue University*

**Reviewed on OpenReview:** *https://openreview.net/forum?id=hwz32S06G4*

## Abstract

Spiking Neural Networks (SNNs) with their bio-inspired Leaky Integrate-and-Fire (LIF) neurons inherently capture temporal information. This makes them well-suited for sequential tasks like processing event-based data from Dynamic Vision Sensors (DVS) and event-based speech tasks. Harnessing the temporal capabilities of SNNs requires mitigating vanishing spikes during training, capturing spatio-temporal patterns and enhancing precise spike timing. To address these challenges, we propose *TSkips*, augmenting SNN architectures with forward and backward skip connections that incorporate explicit temporal delays. These connections capture long-term spatio-temporal dependencies and facilitate better spike flow over long sequences. The introduction of *TSkips* creates a vast search space of possible configurations, encompassing skip positions and time delay values. To efficiently navigate this search space, this work leverages training-free Neural Architecture Search (NAS) to identify optimal network structures and corresponding delays. We demonstrate the effectiveness of our approach on four event-based datasets: DSEC-flow for optical flow estimation, DVS128 Gesture for hand gesture recognition and Spiking Heidelberg Digits (SHD) and Spiking Speech Commands (SSC) for speech recognition. Our method achieves significant improvements across these datasets: up to 18% reduction in Average Endpoint Error (AEE) on DSEC-flow, 8% increase in classification accuracy on DVS128 Gesture, and up to $\sim 8\%$ and $\sim 16\%$ higher classification accuracy on SHD and SSC, respectively.

## 1 Introduction

Spiking Neural Networks (SNNs) are a class of bio-physically realistic models with Leaky Integrate-and-Fire (LIF) (Abbott, 1999) neurons that offer a promising alternative for processing sequential data. By encoding information in the precise timing of spikes, SNNs capture spatio-temporal dependencies (Rathi & Roy, 2020; Chowdhury et al., 2021; Wu et al., 2017) through the membrane potential of their LIF neurons. This intrinsic ability to process temporal information allows SNNs to excel at capturing dynamic patterns, effectively functioning as specialized RNNs (Rathi & Roy, 2024; Deng et al., 2022) with reduced complexity.

The ability of SNNs to leverage the precise timing of spikes offers a distinct advantage over traditional Artificial Neural Networks (ANNs) for sequential tasks. Note, ANNs are "stateless" in nature (Hagenaars et al., 2021b) and their reliance on specially curated input encoding schemes (Zhu et al., 2018b; Paredes-Vallés & De Croon, 2021) hinder their ability to accurately capture precise timing information crucial in event streams, as generated by Dynamic Vision Sensors (DVS) (Lichtsteiner et al., 2008; Brandli et al., 2014). Standard Recurrent Neural Networks (vRNNs) (Rumelhart et al., 1986) and Long Short-Term Memory (LSTM) networks (Hochreiter, 1997; Ponghiran & Roy, 2021; Gehrig et al., 2021b) attempt to address this, but remain difficult to train (Pascanu et al., 2013) and computationally expensive. Furthermore, transformers and spiking transformers (Li et al., 2022a; Zhou et al., 2023; Li et al., 2023) excel at sequential processing but come at the cost of significantly larger models and computational overhead.

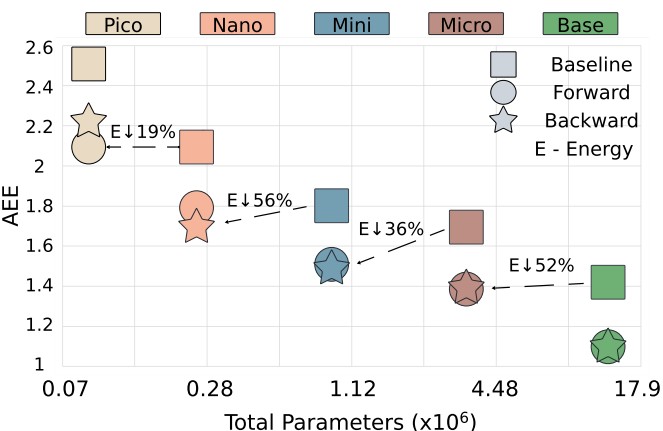

Figure 1: Model size vs. average endpoint error (AEE) for fully spiking EV-FlowNet architectures (Kosta & Roy, 2023) modified with Forward and Backward *TSkips* on the DSEC-flow dataset (lower AEE is better). Each tick on the horizontal axis represents a 3× increase in model size. *TSkips* achieve an average reduction of 15% in AEE and 40.75% in inference energy (E) while maintaining comparable AEE to larger baselines.

Deep SNNs offer the appealing combination of rich temporal information processing and computational efficiency, but training them effectively presents unique challenges (Pfeiffer & Pfeil, 2018; Rathi et al., 2021). The combination of multiple time steps with Back Propagation through Time (BPTT), non-differentiable spikes and sparse event data can worsen the vanishing spike problem (Hagenaars et al., 2021a; Neftci et al., 2019). To circumvent this, previous works have proposed adaptive LIF neurons with trainable threshold and leak (Fang et al., 2021b; Kosta & Roy, 2023; Wang & Li, 2023) and approximate surrogate gradients (Fang et al., 2021a; Neftci et al., 2019). Other works (Shrestha & Orchard, 2018; Sun et al., 2023b; Hammouamri et al., 2023) have considered synaptic delays as an additional parameter alongside weights, thus offering a richer representation of the spike patterns present in event data.

In contrast, this work introduces explicit temporal delays in forward and backward skip connections, named temporal skips (*TSkips*), for SNNs and hybrid ANN-SNN (Kugele et al., 2021; Negi et al., 2024) models. *TSkips* offer finer control over spike timing, enhance responsiveness to temporal patterns, mitigate vanishing spikes, all while capturing long-term spatio-temporal dependencies for event-based data. In addition, our method utilizes adaptive LIF neurons (Yin et al., 2020; Hagenaars et al., 2021a; Kosta & Roy, 2023) that have learnable leaks and thresholds that capture local temporal patterns.

The next question to address would be, what are the optimal *TSkips* configurations within the network architecture and what are the corresponding temporal delays associated with them? To efficiently identify optimal *TSkips* configurations across different architectures and sequence lengths, we leverage training-free Neural Architecture Search (NAS) tailored for SNNs (Kim et al., 2022). *TSkips* introduces minimal increase in model size with very few additional trainable parameters, minimal overhead and improves inference on event-based tasks. In addition, we have observed that training time is faster for the proposed networks compared to standard SNNs, RNNs and LSTMs. The contributions of this work can be summarized as follows:

- We introduce *TSkips*, a novel mechanism that enhances SNNs and hybrid ANN-SNN architectures by enabling direct transmission of spike information between non-adjacent layers with temporal delays, effectively capturing long-term spatio-temporal patterns. (Section 3)

- We analyzed various *TSkips* configurations, revealing the complexities of the search space, particularly the interplay between the temporal delay ($\Delta t$), *TSkips* position and network depth. This

complexity motivated the use of NAS to efficiently identify optimal *TSkips* architectures. (Section 4.2)

- We show that *TSkips* achieves strong performance across sequential tasks. On the DSEC-flow dataset, they reduce average endpoint error (AEE) and inference energy, up to 18% and 56%, respectively, as shown in Fig. 1. Furthermore, *TSkips* achievess significant accuracy improvements on DVS128 Gesture, SHD and SSC datasets — up to 8.3%, 8.6% and 16.04%, respectively. This highlights the scalability and effectiveness of our approach across diverse sequential tasks and architectures. (Section 4)

## 2 Related Work

This section explores the challenges of training deep SNNs and incorporating delays to capture richer temporal dynamics.

### 2.1 Training Deep SNNs

Early methods to train SNNs relied on ANN-to-SNN conversion (Cao et al., 2015; Panda et al., 2020). However, this approach often struggled to match ANN performance on complex sequential tasks (Deng et al., 2020), particularly with event data. To overcome the challenges associated with ANN-to-SNN conversion, Kugele et al. (2020) focus on converting existing neural network architectures into spiking neural networks using streaming rollouts (Fischer et al., 2018). By temporally unfolding the recurrent structures in the spiking domain the DenseNet skip connections essentially have a single step delay, much like vRNNs.

A key obstacle in training deep SNNs is the non-differentiable nature of spike activations (Fang et al., 2021a), which prevents the direct application of gradient-based optimization techniques, leading to subpar performance. To overcome this, approximate surrogate gradients (Fang et al., 2021a; Neftci et al., 2019; Wu et al., 2017) have been proposed, that replace non-differentiable spike activations with smooth gradient approximations, enabling the use of BPTT (Werbos, 1990) to train deep SNNs. However, this approach introduces other challenges like vanishing gradients, especially with long sequences and sparse event data (Hagenaars et al., 2021a; Neftci et al., 2019), and reduce accuracy compared to ANNs (due to approximate gradients). To address these challenges, previous work has investigated two main avenues: adaptive LIF neurons with learnable thresholds and leaks (Fang et al., 2021b; Kosta & Roy, 2023; Wang & Li, 2023), and skip connections in SNNs (Benmeziane et al., 2023). While adaptive LIF neurons allow for dynamic adjustment and better capture of spatio-temporal patterns, skip connections can improve accuracy and efficiency, However, neither approach fully addresses the challenges of representing and capturing complex temporal patterns. To achieve this, we propose *TSkips*, incorporating explicit temporal delays into the network architecture with forward and backward skip connections.

### 2.2 Delays in SNNs

The importance of precise timing information in event-based data, such as that from dynamic vision sensors, has motivated research into effectively capturing temporal dynamics in SNNs (Rathi & Roy, 2020; Chowdhury et al., 2021). SCTT (Soo et al., 2023) explore temporal skips in recurrent networks and are evalauted on cognitive tasks that have long-term temporal dependencies. However, SCTT employs an expensive evaluation strategy to identify optimal delay and temporal skip hyperparameters that involves training several models across multiple tasks to identify optimal connections. TTFS (Kim et al., 2024) explore adding delays in skip connections by encoding the latency of the first spike emitted by a neuron. This encoded delay is then introduced into skip connections in SNNs as learnable parameters that promote local path synchronization. However, the evaluation of both these works are limited to simple tasks that are not typically conducive to SNNs. Furthermore, while the spike encoding strategy of TTFS allows for better synchronization of spikes for the model, it does not account for factors such as noise or irregularities in the event stream or long periods of sparse events.

Other works have introduced delays alongside the weights in SNNs to offer a richer representation of the spike patterns. For instance, Shrestha & Orchard (2018) introduced a fixed delay in SNNs to improve learning of

temporal complexities, Sun et al. (2023b) extended this by making these delays learnable and introducing a layer-specific maximum delay. Hammouamri et al. (2023) further refined this by making the per-layer maximum delay learnable and utilizing Dilated Convolutions with Learnable Spacings (DCLS) (Khalfaoui-Hassani et al., 2023) to capture spatio-temporal patterns. However, learning these delays, as in Sun et al. (2023b); Hammouamri et al. (2023), introduces a new set of parameters with the same dimension as the weights, essentially doubling the number of trainable parameters.

In contrast to these methods that rely on expensive tempral skip configuration evaluation strategies, estimating delay values for local path synchronization or learning delays, our proposed method introduces explicit temporal delays in SNNs. We explore these delays in both forward and backward temporal skip connections, named *TSkips*. The use of *TSkips* coupled with a learnable scaling factor and adaptive LIF neurons effectively captures long-term spatio-temporal patterns with minimal additional computational overhead.

## 3 Methodology

This section details our methodology, starting with the input representation and the neuron model employed in our SNNs, followed by the proposed method, our NAS search space, and the training process.

### 3.1 Input Representation and Sensors

For efficient computation, SNNs require data in the form of discrete spikes, mirroring communication in biological neurons. DVS (Lichtsteiner et al., 2008) cameras output asynchronous spikes representing real-time changes in pixel intensity ($I_t$). This eliminates redundant data transmission and achieves higher temporal resolution than traditional frame-based cameras or rate coding methods (Cao et al., 2015). These cameras trigger events when the log intensity change at a pixel crosses a threshold ($\theta$) (Gallego et al., 2022), represented as $||log(I_t) - log(I_{t-1})|| \geq \theta$. Events are encoded in the Address Event Representation (AER) format as tuples $(x, y, t, p)$ — representing pixel coordinates, timestamp, and polarity. Similarly, event-based audio datasets (Cramer et al., 2022) mimic human auditory spiking activity, with spikes represented as $(x, t)$ — denoting spike unit and timestamp.

### 3.2 Neuron Model

The fundamental computational unit of SNNs is the spiking neuron. We employ Leaky Integrate-and-Fire (LIF) (Abbott, 1999) neurons for their biological plausibility and computational efficiency. The LIF neuron integrates input spike information over time, accumulating it in its membrane potential. This potential gradually decays, allowing controlled forgetting of less relevant information. The dynamics of the LIF neuron can be described by

$$U_l^t = \lambda U_l^{t-1} + W_l O_{l-1}^t - V_l^{th} O_l^{t-1} \tag{1}$$

where $U_l^t$ represents the membrane potential of the neurons in layer $l$ at time step $t$, $\lambda$ is the leak factor, controlling the decay rate of the membrane potential. $W_l$ is the weight matrix for neurons in layer $l$, $V_l^{th}$ is the voltage threshold of layer $l$ and $O_l^t$ is the output generated by layer $l$ at time step $t$.

When the membrane potential surpasses the voltage threshold, the neuron emits a spike. The first term in Eq. (1) denotes the leakage in the membrane potential, the second term computes the weighted summation of output spikes from layer $(l - 1)$ and the third term denotes the reduction in membrane potential when an output spike is generated at layer $l$. The membrane potential is then reset, either to be zero (hard reset) or by subtracting $V_l^{th}$ (soft reset).

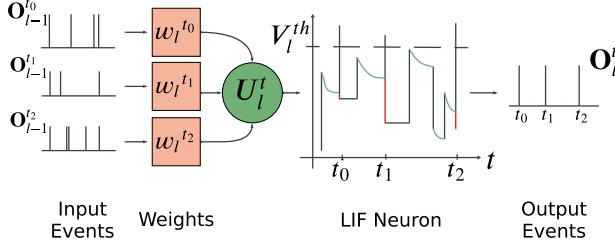

Figure 2: LIF Neuron

Equation (2) describes the spike generation.

$$Z_l^t = \frac{U_l^t}{V_l^{th}} - 1, \quad O_l^t = \begin{cases} 1, & \text{if } Z_l^t > 0 \\ 0, & \text{otherwise} \end{cases} \tag{2}$$

We use a hard reset for optical flow estimation and a soft reset for gesture and speech classification. Fig. 2 shows the LIF neuron dynamics.

### 3.3 Proposed Method

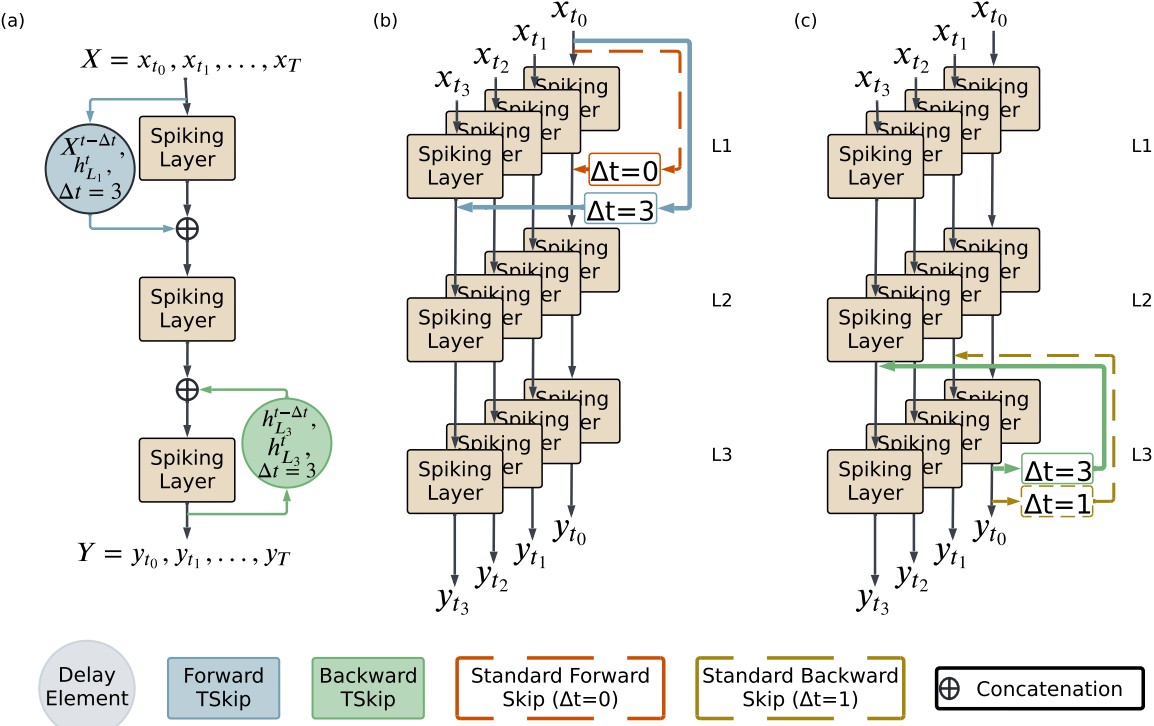

Figure 3: Illustration of *TSkips* in an SNN. (a) A 3-layer SNN with forward and backward *TSkips*, annotated with their origin and destination layers and the delay ($\Delta t$). (b) Unrolled SNN with $T = 4$ with a forward *TSkips* and $\Delta t = 3$ compared with a standard skip connection (no delay). (c) Unrolled SNN with $T = 4$ with a backward *TSkips* and $\Delta t = 3$ compared with a standard backward connection ($\Delta t = 1$) (Rumelhart et al., 1986)

Efficiently training deep SNNs is often hindered by vanishing gradients (Hagenaars et al., 2021a; Neftci et al., 2019), which impede accurate spike propagation across time steps. To address this, we propose a novel method that augments skip connections in SNNs by introducing explicit temporal delays ($\Delta t$) within the connections. We refer to these augmented skip connections, which can be forward, backward, or a combination of both, as *TSkips*. The duration of these explicit delays is constrained by $0 < t - \Delta t < T$, where $t$ is the current time step and $T$ is the sequence length of the data. Specifically, $t$ represents the set of all discrete time steps within the sequence, denoted as $t = \{t_i, \forall i = 0, \cdots, T\}$. This constraint on $\Delta t$ ensures that the network does not receive future information.

To illustrate the information flow in a neural network with *TSkips*, let us consider how they modify the output of a layer. If $h_l^t$ denotes the input to a layer $l$ at time step $t$, then a forward *TSkips* provides input $h_{l-k}^t$ and a backward *TSkips* provides input $h_{l+k}^t$. For brevity, we denote these skip inputs as $h_{l\pm k}^t$, where $k$ indicates the layer separation between layer $l$ and the source of the skip connection. *TSkips* can be represented as:

$$h_l^t = f_l(h_{l-1}^t \oplus W_s h_{l\pm k}^{(t-\Delta t)}) \tag{3}$$

where $f_l$ is the affine function of layer $l$, $\oplus$ represents either concatenation (Huang et al., 2017) or element-wise addition (He et al., 2015) and $W_s$ is a fixed shortcut path matrix (He et al., 2016) that randomly selects channels in $h_{l\pm k}^{(t-\Delta t)}$ to match the dimensions of $h_l^t$. Eq. 3 demonstrates that the output of layer $l$ at time step $t$ is now influenced by the output of layer $l \pm k$ at time step $(t - \Delta t)$. Here, $k$ represents the difference in layer indices between connected layers. For example, if $l = 4$ and $k = 2$, a forward *TSkips* connects layers 2 and 4 with delay $\Delta t$.

Our method, incorporating *TSkips*, is illustrated in Fig. 3. Fig. 3(a) shows a simple 3-layer SNN architecture, illustrating both forward and backward *TSkips* between adjacent layers. Each connection is annotated with the origin and destination layers and ($\Delta t$). Fig. 3(b) depicts the temporal unrolling of the network over $T = 4$ time steps. It contrasts a standard skip connection ($\Delta t = 0$) and a forward *TSkips* with $\Delta t = 3$. Fig. 3(c) similarly illustrates a backward *TSkips* with $\Delta t = 3$ and contrasts it with a standard backward connection that has a delay of $\Delta t = 1$, typically seen in vRNNs (Rumelhart et al., 1986). This comparison emphasizes that *TSkips* allow for more flexibility and can thus capture longer temporal dependencies compared to standard RNNs. This flexibility comes with minimal overhead, as *TSkips* are identity mappings that introduces very few additional parameters and no computational complexity to the network.

*TSkips* enhance the ability of SNNs and hybrid models to process sparse event streams by facilitating the propagation of temporally relevant information across non-adjacent layers. By capturing these long-term dependencies, *TSkips* can improve the network's ability to learn complex temporal patterns. Furthermore, the explicit temporal delays in *TSkips* offer finer control over spike timing within the network. This finer control enhances the network's responsiveness to temporal patterns by allowing it to selectively integrate information from different time steps. Additionally, by enabling the network to access relevant information from the past, *TSkips* can mitigate the issue of vanishing spikes. To further enhance temporal representation, we introduce a learnable scaling factor, $\alpha$, that controls the ratio of data at the current time-step ($t$) and data from the delayed time step ($t - \Delta t$) within the *TSkips*. This is incorporated into the skip as shown in Eq. 4:

$$h_l^t = f_l(h_{l-1}^t \oplus W_s(\alpha h_{l\pm k}^t + (1-\alpha) h_{l\pm k}^{(t-\Delta t)})) \tag{4}$$

### 3.4 Exploring the search space

Identifying optimal models with *TSkips* is challenging due to the vast number of potential architectures and associated *TSkips* configurations. To manage this complexity, we initialize our search with a backbone network (Cai et al., 2020). This backbone provides a starting point for exploration, reducing the number of possible configurations and allowing the search to focus on optimizing the specific parameters of our proposed method. However, significant complexity remains due to the numerous choices for $\Delta t$, connection placement, and network depth.

To illustrate this complexity, consider a 3-layer network with the possibility of forward and backward connections between any two layers (Fig. 4). This simple example results in 12 possible combinations of skip connections. With a sequence length of $T = 6$, this grows to 120 possible delay combinations and $2^{120}$ total possible *TSkips* configurations. This combinatorial problem, grows exponentially with the sequence lengths, as each additional time step introduces new *TSkips* configurations.

This large search space necessitates an efficient exploration method to identify optimal architectures. To ad-

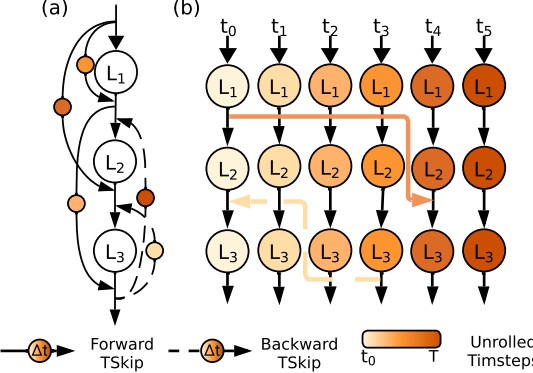

Figure 4: Illustration of the NAS Search Space. (a) Example of possible *TSkips* combinations in a 3-layer network with $T = 6$. (b) Top-ranked forward and backward *TSkips* with $\Delta t = 4$ and $\Delta t = 3$, respectively, found by (Kim et al., 2022). For visual clarity, the figure depicts the *TSkips* spanning a single delay window ($\Delta t$). Here $t_0$ refers to the first time step and $T$ is the sequence length.

dress this, we leverage NASWOT-SAHD (Kim et al., 2022), a training-free NAS method tailored for SNNs. NASWOT-SAHD scores networks at initialization using the Sparsity Aware Hamming Distance (SAHD), which quantifies the dissimilarity in spike patterns between different layers. By favoring networks with diverse spiking activity, SAHD acts as a proxy for identifying optimal sub networks of the backbone architecture augmented with *TSkips*. Specifically, (Kim et al., 2022) performs a random search for architectures with optimal $\Delta t$, *TSkips* positions (origin and destination) and network depth.

### 3.5 Training with Explicit Temporal Delays

Training SNNs and hybrid models with gradient descent requires addressing the non-differentiable nature of spike activations (Fang et al., 2021a) in LIF neurons. We employ adaptive LIF neurons (Kosta & Roy, 2023) with learnable leaks and thresholds, and use the ArcTangent (Fang et al., 2021a) surrogate gradient to approximate the derivative of the spike activation function. To capture precise spike timing and spatio-temporal dependencies, we unroll the network in time. This, combined with adaptive LIF neurons and *TSkips*, allows for explicit representation of the network's temporal evolution, capturing both long-term and local temporal patterns. We then use back propagation through time (BPTT) (Werbos, 1990) for gradient propagation and Batch Normalization Through Time (BNTT) (Kim & Panda, 2021) for improved training stability.

## 4 Experiments

### 4.1 Experimental Setup

**DSEC-Flow**    The DSEC-Flow dataset (Gehrig et al., 2021a) consists of 24 challenging driving sequences with varying lighting conditions, fast motion, and occlusions. To assess our models on unseen data, we created a custom test set by splitting the training set, following the approach in Ponghiran et al. (2023). To evaluate the accuracy of the predicted optical flow, we use average end point error (AEE) (Zhu et al., 2018a), which measures the average Euclidean distance between the predicted and ground truth flow vectors.

$$AEE = \frac{1}{n} \sum_n \|(u,v)_{pred} - (u,v)_{gt}\|_2 \tag{5}$$

For experiments on DSEC-flow (Gehrig et al., 2021a), we use a fully spiking (Kosta & Roy, 2023) and hybrid (Negi et al., 2024) multi-scale encoder-decoder network inspired by EV-FlowNet (Zhu et al., 2018a) as our backbone architecture. We augment this architecture by integrating *TSkips* into the existing skip connections between the encoder and decoder blocks, as depicted in Fig. 5. Due to the inherent structure of our backbone, the search space for optimal *TSkips* configurations is relatively small, focusing primarily on selecting which existing skip connection to modify and determining the appropriate temporal delay ($\Delta t$).

**DVS128-Gesture**    The IBM DVS128 Gesture dataset (Hu et al., 2022) contains 1342 instances of 11 hand gestures, captured with a DVS128 (Lichtsteiner et al., 2008) camera. For experiments on DVS128 Gesture, we use a ResNet18 (He et al., 2015) backbone architecture (Cai et al., 2020) and explore various configurations by modifying its basic blocks. This search includes modifying the input and output channels of each layer, kernel sizes, stride, and network depth. To integrate *TSkips* in the backbone, we remove the original ResNet18 skip connections and search for optimal *TSkips* placement and corresponding $\Delta t$.

For optical flow estimation and gesture recognition, we introduce a learnable scaling factor, $\alpha$, within the *TSkips* to further enhance temporal processing (detailed in Section. 3.3).

**SHD and SSC**    The SHD (Cramer et al., 2022) dataset comprises 10,000 recordings of spoken digits (zero to nine in English and German), while the larger SSC (Cramer et al., 2022) dataset contains 100,000 recordings of spoken words from various speakers. These datasets present challenges for audio classification, requiring the network to capture and process subtle temporal patterns within spike trains that encode spoken sounds. We focus on the top-one classification task for all 35 classes in SSC and all 20 classes in SHD. For our experiments on speech recognition, we use a fully connected Multi Layer Perceptron(MLP) (Rumelhart et al., 1986) as our backbone architecture. We explore various MLP configurations by adjusting the input

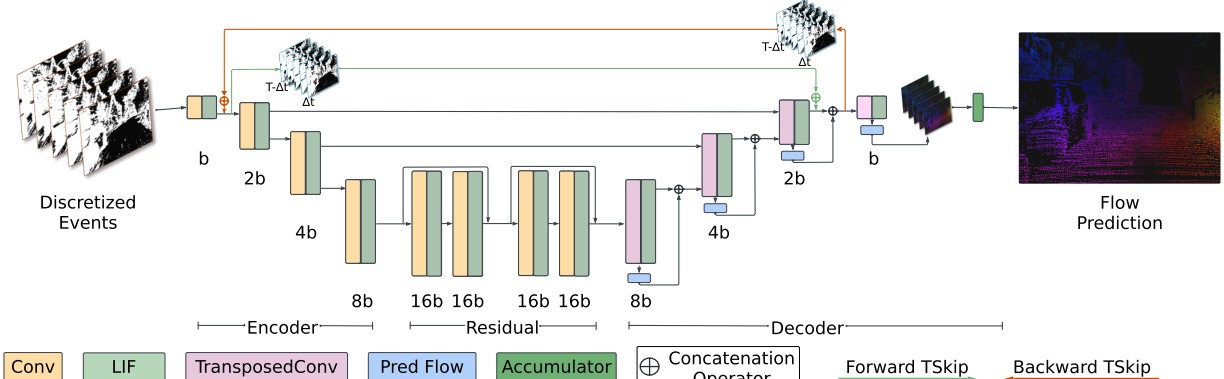

Figure 5: Fully spiking EV-FlowNet (Zhu et al., 2018a) architecture with forward and backward *TSkips*. Each *TSkips* replaces a pre-existing skip connection between the encoder and decoder. Although shown together for conciseness, forward and backward connections are evaluated independently. In practice, when used together, they connect different encoder-decoder layers. This architecture can be modified to be a hybrid ANN-SNN model, by replacing the LIF neurons with ReLU activations.

and output features of each layer, the overall network depth, and the placement and $\Delta t$ of *TSkips*. In these experiments, we omit $\alpha$ used in the EV-FlowNet and DVS128 Gesture *TSkips* architectures.

To improve generalization, we augment the training sets for DVS128 Gesture, SHD and SSC with channel jitter and random noise (Shen et al., 2023). We use sequence lengths (T) of 10, 30, and 99 for DSEC-flow, DVS128 Gesture, and SHD/SSC, respectively. All models are trained on an NVIDIA A40 GPU using the Adam optimizer (Kingma & Ba, 2014). For DSEC-flow, we use a multi-step learning rate scheduler (initial rate: $10^{-3}$, scaled by 0.7 every 10 epochs) and the supervised mean squared error (MSE) loss (Kosta & Roy, 2023; Negi et al., 2024) for 200 epochs. For all other datasets, we use a Cosine Annealing Learning Rate Scheduler (Loshchilov & Hutter, 2017) (initial rate: 0.001, minimum rate: $5 \times 10^{-6}$, updated every 10 iterations) for 100 epochs.

Details on the constraints on the search parameters and *TSkips* configurations for DSEC-flow, DVS128 Gesture, SHD, and SSC can be found in Appendix D.1.

### 4.2 Ablation Study

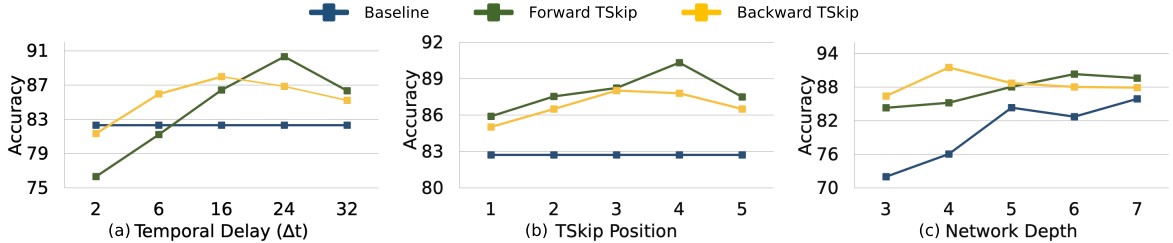

Figure 6: Ablation study results on the SHD dataset demonstrating the impact of varying the (a) temporal delay ($\Delta t$), (b) *TSkips* position and (c) network depth on classification accuracy.

As detailed in Section 3.4, the search space of possible *TSkips* configurations is exponentially large. To effectively navigate this complex search space, we performed an ablation study on an 8-layer MLP baseline found using Kim et al. (2022) on the SHD dataset. We varied the temporal delay ($\Delta t$), *TSkips* position and network depth using concatenation-based (Huang et al., 2017) *TSkips* for our analysis (see Appendix C.6 for detailed baseline network configuration).

**Impact of Temporal Delay ($\Delta$t):** We investigated the impact of varying $\Delta t$ for both forward and backward *TSkips* (Fig. 6(a)). Forward connections connected the input data to the second layer, while backward connections connected the final output to the seventh layer, with a delay of $\Delta t$. Varying $\Delta t$ significantly influenced accuracy, with larger delays generally leading to better classification. This highlights the importance of optimizing $\Delta t$ relative to the sequence length ($T$), as both excessively small and large delays can hinder performance.

**Impact of TSkips Position:** Next, we investigated the impact of *TSkips* position on classification accuracy (Fig. 6(b)). Fixing $\Delta t = 16$, we varied the destination layer of both forward and backward connections. Forward connections originated from the input data, while backward connections originated from the final output. Our analysis revealed that connecting *TSkips* to deeper layers consistently improved accuracy. This suggests that propagating temporal information to deeper layers is crucial for enhancing model performance. Note, the optimal placement varies based on forward or backward *TSkips*.

**Impact of Network Depth:** We analyzed the impact of *TSkips* on models of varying depth (Fig. 6(c)), with forward connections originating from the input and backward connections originating from the final output with $\Delta t = 16$. Our analysis revealed that deeper networks benefited more from forward connections, while shallower architectures favored backward connections. This suggests the optimal configuration of *TSkips* and network depth are intertwined.

These findings underscore the complex interplay between $\Delta t$, *TSkips* position (origin and destination layers), and network depth, which we set as our search parameters. For the results presented in the following sections, we leveraged Kim et al. (2022) to identify optimal baseline and concatenation-based (Huang et al., 2017) *TSkips* architectures (see Appendix C.5 for comparisons with addition-based *TSkips*), constraining the architecture search by the number of parameters.

## 4.3 DSEC-Flow Results



(a) GT      (b) Mini - Baseline      (c) Mini - **F** *TSkip*      (d) Mini - **B** *TSkip*      (e) Base - Baseline

Figure 7: Qualitative results on the DSEC-flow dataset. (a) Ground Truth (GT) mask, (b) Mini (3.4M) SNN baseline model mask, (c) Mini SNN (3.4M) with forward (F) TSkips, (d) Mini SNN (3.4M) with backward (B) TSkips, and (e) Base (13M) SNN baseline model mask. As observed, the Mini model that is 3.8× smaller than Base model performs just as well qualitatively when *TSkips* are incorporated.

Incorporating *TSkips* into the fully spiking and hybrid EV-FlowNet architectures (Kosta & Roy, 2023; Negi et al., 2024), as detailed in Section 4.1, significantly lowers AEE on the DSEC-flow dataset (Table 1). Across both fully spiking and hybrid architectures, *TSkips* consistently yielded an average reduction in AEE of 14% and 9.5%, respectively. Notably, these improvements are achieved with models 3× smaller than baselines with comparable AEE. Although SOTA methods such as E-RAFT (Gehrig et al., 2021b) and E-FlowFormer (Li et al., 2023) achieve 6% and 10% lower AEE , respectively, these reductions come at the cost of increased complexity. E-RAFT, for example, while being 3× smaller than our best model, relies on complex processing of event data before its gated recurrent unit (GRU) update. These processing operations involve converting each event to a voxel grid, extracting features using CNNs and building a correlation volume. These dense kernel operations are computationally expensive during both training and inference, as they must be performed for every event sample. E-FlowFormer is a transformer-based architecture which relies on dense MAC operations in it's self and cross attention blocks. These event processing steps lead to higher inference energy compared to *TSkips* architectures, which take advantage of the inherent sparsity and event-driven nature of SNNs, performing computationally cheaper sparse AC operations. Additionally, E-FlowFormer is trained on a large custom dataset and DSEC-flow, which introduces a significant increase

in computation power and memory during training. Fig. 7 provides a qualitative analysis of how the $3\times$ larger baselines perform compared to the smaller *TSkips* variants which highlights the efficiency of *TSkips* in allowing accurate temporal processing without increasing model complexity. Hyper parameters and a detailed analysis of convergence and inference energy for *TSkips* on DSEC-flow models are in Appendix B.

Table 1: AEE on DSEC-flow (lower is better). Best performing SNN and hybrid baselines and forward/backward *TSkips* models are highlighted in bold.

| Architecture | #Params (M) | AEE | |
| --- | --- | --- | --- |
| | | SNN | Hybrid |
| EV-FlowNet (Gehrig et al., 2021b) | 13.04 | 2.32 | - |
| LSTM-FlowNet (Ponghiran et al., 2023) | N/A[1] | 1.28 | - |
| E-RAFT (Gehrig et al., 2021b) | 5.3 | 0.79 | - |
| E-FlowFormer (Li et al., 2023) | N/A | **0.76** | - |
| *Our Models* | | | |
| Base - Baseline | 13.04 | **1.35** | **0.96** |
| Base + Forward *TSkips* | | **1.12** | 0.86 |
| Base + Backward *TSkips* | | 1.13 | **0.84** |
| Mini - Baseline | 3.41 | 1.65 | 1.11 |
| Mini + Forward *TSkips* | | 1.44 | 1.02 |
| Mini + Backward *TSkips* | | 1.46 | 1.01 |
| Micro - Baseline | 0.93 | 1.80 | 1.22 |
| Micro + Forward *TSkips* | | 1.57 | 1.12 |
| Micro + Backward *TSkips* | | 1.56 | 1.09 |
| Nano - Baseline | 0.27 | 2.17 | 1.47 |
| Nano + Forward *TSkips* | | 1.86 | 1.37 |
| Nano + Backward *TSkips* | | 1.77 | 1.35 |
| Pico - Baseline | 0.092 | 2.57 | 2.02 |
| Pico + Forward *TSkips* | | 2.19 | 1.78 |
| Pico + Backward *TSkips* | | 2.28 | 1.81 |

### 4.4 DVS128-Gesture, SHD and SSC Results

Using *TSkips* in ResNet18 and MLP backbone architectures, as detailed in Section 4.1, consistently improved the classification accuracy of SNNs across the DVS Gesture, SHD, and SSC datasets (Tables 2 and 3).

On the DVS Gesture dataset, incorporating *TSkips* resulted in a significant 8.37% improvement in accuracy over the baseline model, a convolutional SNN found by Kim et al. (2022). Importantly, our approach achieved this with significantly smaller models compared to SOTA, including a spiking transformer (Qin & Liu, 2024), demonstrating the efficiency of our proposed method. Furthermore, our method outperformed several other approaches, including spiking RNNs (Xing et al., 2020), methods incorporating learnable delays (Shrestha & Orchard, 2018), and methods using specialized LIF neurons or spatio-temporal feature extraction (Jiang & Zhang, 2024; Samadzadeh et al., 2023). This highlights the effectiveness of our approach in capturing and leveraging temporal dependencies for improved gesture recognition.

On the SHD dataset, *TSkips* achieved an average accuracy gain of 8.6% over the 4-layer baseline (Baseline - 1) and 8.31% over the 8-layer baseline (Baseline - 2). Our approach also outperformed various recurrent architectures (vRNNs, spiking RNNs, and LSTMs) by substantial margins. Similarly, on the SSC dataset, *TSkips* yielded an average accuracy gain of 14% over Baseline - 1 and 16.04% over the 8-layer Baseline - 2, again surpassing the performance of recurrent architectures.

---

[1]N/A - not reported #Params

[2]Backward skips are added to all baseline layers with $\Delta t = 1$.

[3]Zhou et al. (2023) was evaluated on SHD and SSC with minimal updates to the architecture and tuned learning rate.

Table 2: Classification accuracy on DVS-Gesture. Comparisons between forward/backward *TSkips* and SOTA models. Best models in bold.

| Method | Base Model | #Params (M) | Accuracy (%) |
|---|---|---|---|
| SLAYER (Shrestha & Orchard, 2018) | SNN (8 layer) | N/A | 93.6 |
| KLIF (Jiang & Zhang, 2024) | VGG-11 | N/A | 93.75 |
| DECOLE (Kaiser et al., 2020) | Custom | N/A | 95.2 |
| SNN-Skip (Benmeziane et al., 2023) | ResNet18 | N/A | 95.43 |
| Streaming rollout SNN Kugele et al. (2021) | DenseNet | 0.8 | 95.56 |
| SCRNN (Xing et al., 2020) | Custom | N/A | 96.59 |
| STS-ResNet (Samadzadeh et al., 2023) | ResNet18 | N/A | 96.7 |
| Mamba-Spike (Qin & Liu, 2024) | Mambda | 6.1 | 97.8 |
| *Our Models* | | | |
| Baseline | | 0.30 | **88.75** |
| vRNN[2] | ResNet18 | 0.30 | 89.12 |
| Forward *TSkips* | Backbone | 0.42 | 94.82 |
| Backward *TSkips* | | 0.38 | 95.97 |
| Forward + Backward *TSkips* | | 0.55 | **97.52** |

Table 3: Classification accuracy on SHD and SSC datasets. Comparisons between forward, backward and a combination of both *TSkips* against SOTA models and two baseline models: a shallow 4-layer MLP network (Baseline - 1) and a deep 8-layer MLP (Baseline - 2). Best models with a single and two *TSkips* in bold.

| Method | #Params (M) | | Accuracy (%) | |
|---|---|---|---|---|
| | SHD | SSC | SHD | SSC |
| *SOTA Models* | | | | |
| LSTM-LIF (Zhang et al., 2023) | 0.14 | 0.11 | 88.91 | 63.46 |
| SRNN (Yin et al., 2021) | N/A | N/A | 90.4 | 74.2 |
| DL256-SNN-DLoss (Sun et al., 2023a) | 0.14 | - | 92.56 | - |
| SpikGRU (Dampfhoffer et al., 2022) | - | 0.28 | - | 77.00 |
| radLIF (Bittar & Garner, 2022) | 3.9 | 3.9 | 94.62 | 77.40 |
| Spikingformer (Zhou et al., 2023)[3] | 1.98 | 1.99 | 82.68 | 72.43 |
| DCLS-Delays (Sun et al., 2023b) | 0.21 | 2.5 | **95.07** | **80.69** |
| *Our Baseline MLPs* | | | | |
| Baseline - 1 | 0.16 | 0.12 | 84.32 | 64.19 |
| Baseline - 2 | 1.04 | 1.04 | **86.42** | **67.54** |
| *Our MLP Models with TSkips* | | | | |
| vRNN[2] - 1 | 0.16 | 0.12 | 68.50 | 71.20 |
| Forward *TSkips* - 1 | 0.24 | 0.42 | 92.32 | 76.50 |
| Backward *TSkips* - 1 | 0.19 | 0.24 | 93.64 | **79.87** |
| Forward + Backward *TSkips* - 1 | 0.20 | 0.54 | 93.01 | 78.64 |
| vRNN[2] - 2 | 1.04 | 1.04 | 71.35 | 72.24 |
| Forward *TSkips* - 2 | 1.12 | 1.08 | **94.15** | 78.98 |
| Backward *TSkips* - 2 | 1.16 | 1.14 | 93.86 | 79.65 |
| Forward + Backward *TSkips* - 2 | 1.28 | 1.42 | **94.73** | **80.23** |

While *TSkips* achieves accuracy within 0.34% of SOTA model DCLS-Delays (Hammouamri et al., 2023) on SHD and 0.46% on SSC, it offers several distinct advantages. Specifically, the architectural constraint of DCLS-Delays to feedforward networks highlights a key area where *TSkips* provides a complementary alternative. Furthermore, *TSkips* as an architectural addition coupled with NAS allows for modifications

to various network architectures, as shown in Section 4.3 and also offers efficiency due to very few new trainable parameters coupled with strong performance. In particular *TSkips* achieves accuracy comparable to Hammouamri et al. (2023) on the SSC dataset with a 44% smaller model.

These improvements can be attributed to the ability of *TSkips* to capture long-term temporal dependencies, facilitated by the unrolled network structure, detailed in Section 3. Our method improves both the accuracy and convergence rate of the models by mitigating vanishing spikes and enabling more efficient gradient propagation during training. The analysis presented in Appendices A and C demonstrates lower energy consumption and faster convergence compared to standard SNNs and recurrent architectures. Additionally, we show that incorporating *TSkips* in ANNs and hybrid models significantly improves speech classification (Appendix C.4).

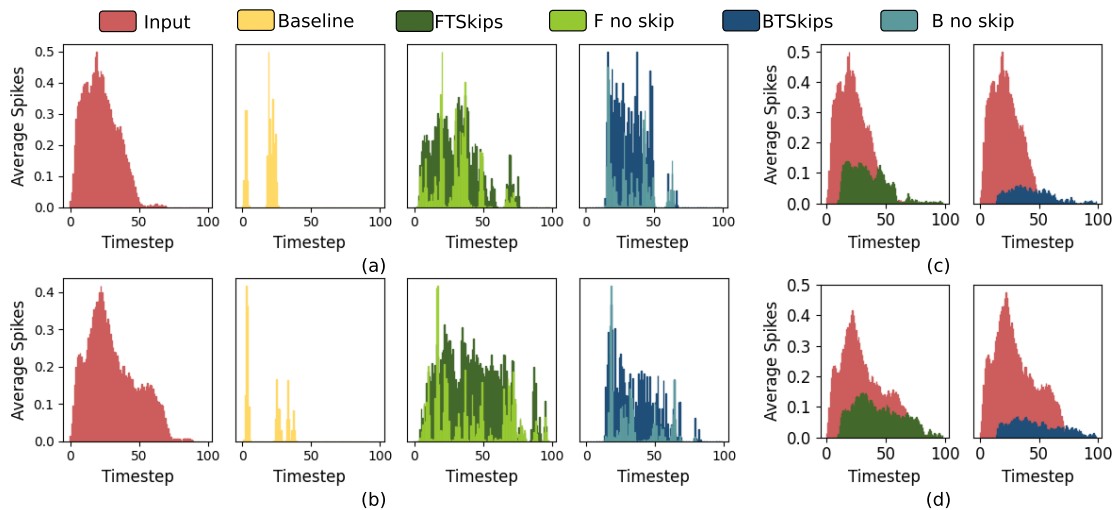

Figure 8: Trained network's response to temporal patterns at inference, visualized by average spiking activity. For two randomly sampled SHD event streams ((a) and (b)), we compare the true event activity against that of a baseline MLP, a forward *TSkips* (F *TSkips*) network, the F *TSkips* network with the skip removed (F no skip), a backward *TSkips* (B *TSkips*) network, and the B *TSkips* network with the skip removed (B no skip). Sub-figures (c) and (d) show the average spiking activity specifically within the forward and backward *TSkips*, demonstrating spike generation aligned with the specified delay window. All plots are normalized by the 700-dimensional input feature representation.

### 4.5 Observations

To empirically validate our claim that *TSkips* enhance the network's responsiveness to temporal patterns, we analyze the spike representations of trained models at inference. Fig. 8 compares the average spiking activity of trained models with and without *TSkips* against the true event stream for two randomly sampled SHD data points. The *TSkips* models without a temporal skip have a zero tensor passing through the *TSkips* connection to ensure the model being evaluated does not undergo any modifications. Figures 8(a) and 8(b) show that the average spiking pattern of the trained models (even after removing the *TSkips*, called F/B no skip) aligns more closely with the ground truth average spike distribution compared to the baseline network. Removing *TSkips* during inference reduces the average spike rate and memory requirements, and the qualitative results validate that the *TSkips* model has learned the input event distribution and captures the input dynamics better than the baseline. We attribute this to *TSkips* aiding the adaptive LIF parameters and weights to learn more effectively during BPTT.

Furthermore, Fig. 8(c) and (d) isolate the average spiking activity within the forward (F) and backward (B) *TSkips* connections themselves. These plots reveal two key observations: first, spike generation within the *TSkips* aligns with the specified temporal delay ($\Delta t$), and second, the activity mirrors a scaled version of the original input distribution. This provides strong empirical evidence that *TSkips* propagates relevant temporal information from the past. Furthermore, Figs. 8(a) and (b) validate that *TSkips* improve learning rather than acting as additional connections. The distinct characteristics of F and B *TSkips* are also evident.

F *TSkips* exhibit more distributed spike patterns and higher variability, likely due to processing denser latent representations from earlier network stages. The smaller and sparser activity in B *TSkips* might facilitate more nuanced error correction during BPTT. Crucially, both types of *TSkips* demonstrate sensitivity to the entire event stream, including the sparse tail end, as seen in Figures 8(a) and (b). The sustained activity in Fig.8(a) and (b) during the later, sparser parts of the sequence in both ( *TSkips*/no *TSkips*) model evaluations at inference indicates integration of information from earlier, denser time steps.

This ability of *TSkips* to integrate information across different temporal contexts allows the network to build richer latent representations. This mechanism is particularly important for mitigating vanishing gradients/spikes during BPTT, as *TSkips* establish direct pathways for information and gradients across long temporal distances, ensuring learning even when events are sparser over the sequence length. We provide quantitative results on these 'no skip' models for the DSEc-flow and SHD datasets in Appendixes B.2 and C.2.

### 4.6 Scalability of *TSkips*

To assess the scalability of *TSkips* to deeper architectures and more complex visual tasks, we evaluated our method on the CIFAR10-DVS (Li et al., 2017) dataset. While the DSEC-flow results (Section 4.3) demonstrated the effectiveness of *TSkips* on a complex real-world task, the underlying EV-FlowNet architecture is custom-designed. To further validate that *TSkips* works on standard, widely adopted architectures in a more complex event-based setting, we applied our method to larger models like spiking ResNet18 and VGG11, incorporating the NDA data augmentation technique (Li et al., 2022b) as our baseline. The results presented in Table 4 consistently demonstrate that *TSkips* improve accuracy over baseline models in these deeper networks.

Table 4: Classification accuracy on the CIFAR10-DVS dataset, comparing the performance of baseline spiking ResNet18 and VGG11 models against their *TSkips*-augmented counterparts, illustrating the scalability of our approach. We use the NDA (Li et al., 2022b) spiking ResNet18 and VGG11 models as our baselines.

| Method | Base Model | #Params (M) | Accuracy (%) |
|---|---|---|---|
| SOTA Models | | | |
| Spikingformer (Zhou et al., 2023) | Single-stage ViT | 2.57 | 81.3 |
| $\alpha$-SSA (Xiao et al., 2025) | Multi-stage ViT | 1.54 | 82.24 |
| SpikingResformer (Shi et al., 2024) | ResNet+ViT | 35.52 | **84.8** |
| Basline Models | | | |
| Baseline - 1 (NDA (Li et al., 2022b)) | ResNet18 | 11.700 | 78.00 |
| Baseline - 2 (NDA (Li et al., 2022b)) | VGG11 | 132.86 | 81.70 |
| *TSkips* Models | | | |
| F *TSkips* - 1 | ResNet18 | 11.708 | 81.08 |
| B *TSkips* - 1 | ResNet18 | 12.224 | **82.93** |
| F *TSkips* - 2 | VGG11 | 134.04 | 82.56 |
| B *TSkips* - 2 | VGG11 | 134.04 | **83.01** |

The CIFAR10-DVS dataset (Li et al., 2017) provides a valuable test case, showing that *TSkips* can provide accuracy gains even for event data with artificially generated temporal information from repeated closed-loop movements. These results offer insight into the effectiveness of *TSkips* in challenging vision/audio tasks that possess rich temporal dynamics, as detailed in Sections 4.3 and 4.4, demonstrating their ability to improve

performance in deeper large-scale networks on event data. However, it is important to consider that smaller transformer models, such as Zhou et al. (2023); Xiao et al. (2025); Shi et al. (2024) can achieve similar or better performance with significantly fewer parameters.

Beyond specific task performance, the vast search space for optimal *TSkips* configurations (temporal delay ($\Delta t$) and skip positions) is a recognized challenge. This motivated our use of training-free NAS for efficient exploration. As detailed in Section 3.4, the NASWOT-SAHD (Kim et al., 2022) proxy inherently guides this search by favoring networks with high linear separability in LIF activations across time steps, suggesting robustness beyond just favorable initializations. To confirm that *TSkips* consistently provide performance improvements over strong baselines identified by NAS, we analyzed the best-found baseline and their *TSkips*-augmented variants across three random seeds. These results for DSEC-flow (Gehrig et al., 2021a) are provided in Appendix D.2, Table 12, further validating the consistent improvement in inference with *TSkips*.

## 5    Discussion

While our approach demonstrates promising results, several avenues for future investigation remain. Enhancing the energy efficiency of discovered architectures is one key area, potentially through energy-based regularization within a training-free NAS proxy. Furthermore, mitigating the increase in memory usage associated with *TSkips* and temporal unrolling requires exploring memory-efficient training techniques or alternative architectures. Future research could also investigate making the temporal delays learnable/ dynamically changing and inference energy optimization techniques to further refine the efficiency of our method.

Implementing *TSkips* in neuromorphic or biological systems presents distinct challenges. The storage of historical neural states to enable connections across time steps necessitates memory allocation, which can be a significant constraint on neuromorphic hardware with limited on-chip resources. *TSkips* can be implemented on standard hardware by storing and retrieving past layer states, incurring memory and energy costs for simulation. The analysis in Sec. 4.5 provides a promising solution to this problem, where the trained models can be deployed without hidden state storage and passing a null input through the *TSkips*. On neuromorphic hardware, *TSkips* can be realized through direct routing with delays, where information from an earlier layer is directly routed to a later layer with a controlled temporal delay. This can be achieved using physical delay lines or digital buffers within the chip's architecture, potentially offering energy efficiency by bypassing computations in intermediate layers. Importantly, our findings in Section 5 show that trained *TSkips* models can perform well even without explicitly using the temporal skip at inference, reducing hardware memory and energy requirements. Biologically, analogies might be found in short-term memory mechanisms or axonal delays that could bridge temporal gaps in information flow. Overcoming connectivity and communication latency limitations in neuromorphic hardware will also be crucial for realizing flexible temporal skips. Balancing the performance gains of *TSkips* with these hardware costs will guide future work exploring memory-efficient encoding of historical states, incorporating hardware constraints into the NAS process, and investigating how to mitigate the inherent hardware delays using *TSkips*.

## 6    Conclusion

In conclusion, this work demonstrates the potential of incorporating temporal delays in forward and backward skip connections across SNNs and hybrid architectures. Our proposed method, optimized through training-free NAS, enhances accuracy, temporal representation, and efficiency while maintaining small model sizes, as shown by our results on the DSEC-flow, DVS128 Gesture, SHD, and SSC datasets. Furthermore, our ablation studies reveal the importance of understanding the interplay between network depth, temporal delays, and connection placement in optimizing SNN performance. Overall, this work demonstrates a promising direction for the development of efficient and accurate SNN architectures, achieving competitive performance with reduced complexity, thus offering a compelling alternative for real-world applications requiring fast and accurate processing of temporal information.

## Acknowledgment

This work was supported by, the Center for the Co-Design of Cognitive Systems (COCOSYS), a DARPA-sponsored JUMP 2.0 center, the Semiconductor Research Corporation (SRC), and the National Science Foundation.

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

# A    Inference Energy Estimation

To validate the observation that *TSkips* exhibit reduced energy consumption compared to baseline SNNs and standard recurrent methods (vRNNs (Rumelhart et al., 1986) and LSTMs (Hochreiter, 1997)) for various tasks (Sections 4.3 and 4.4), we estimate inference energy. These estimations are made using the method from Rueckauer et al. (2017); Lee et al. (2021), which considers the energy required for sparse accumulate (AC) operations in SNNs ($E_{AC} = 0.9pJ$) and dense multiply-and-accumulate (MAC) operations in ANNs ($E_{MAC} = 4.6pJ$) based on a 45nm CMOS technology (Horowitz, 2014). We calculated the total inference energy ($E_{total}$) as:

$$E_{total} = \begin{cases} \#OPS_{SNN} \times E_{AC}, & \text{for SNNs} \\ \#OPS_{ANN} \times E_{MAC}, & \text{for ANNs} \end{cases} \qquad (6)$$

where $\#OPS_{SNN}$ and $\#OPS_{ANN}$ represent the layer-wise synaptic connections in SNNs and ANNs, respectively, calculated as:

$$\#OPS_{SNN} = T \times \mathcal{N} \times \mathcal{C} \times \mathcal{M},$$
$$\#OPS_{ANN} = \mathcal{N} \times \mathcal{C}.$$

Here, $T$ is the sequence length, $\mathcal{N}$ is the number of neurons in the layer, $\mathcal{C}$ is the number of synaptic connections per neuron, and $\mathcal{M}$ is the average spike rate over $T$.

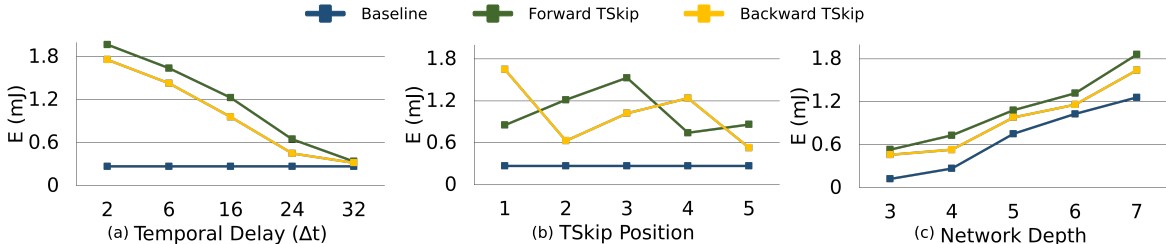

Figure 9: Ablation study on the SHD dataset demonstrating the impact of varying the (a) temporal delay ($\Delta t$), (b) *TSkips* position and (c) network depth on inference energy.

### A.1 Inference Energy Ablation Study

Building upon our ablation study in Section 4.2, we further investigated the impact of different *TSkips* configurations on inference energy, estimated using Eq. 6. Specifically, we explored the impact of varying temporal delay ($\Delta t$), *TSkips* position, and network depth on an 8-layer multilayer perceptron (MLP) baseline model with concatenation-based *TSkips*, found using the Neural Architecture Search (NAS) framework (Kim et al., 2022). This analysis is crucial because *TSkips* increases the average spike rate within the network, potentially leading to higher inference energy consumption.

**Impact of Temporal Delay ($\Delta t$):** We investigated the impact of varying $\Delta t$ for both forward and backward *TSkips* on inference energy (Fig. 9(a)). Forward connections originated from the input data and terminated at the second layer, while backward connections originated from the final output and terminated at the seventh layer. Our analysis revealed that varying $\Delta t$ significantly influenced inference energy. Larger delays resulted in lower energy consumption, likely because fewer (but more temporally relevant) spikes were propagated through the connections.

**Impact of TSkip Position:** Next, we investigated the impact of varying *TSkips* placement on inference energy (Fig. 9(b)). With a fixed $\Delta t = 16$, we varied the destination layer of both forward (originating from the input data) and backward (originating from the final output) connections. While no clear trend emerged, terminating both forward and backward *TSkips* at deeper layers generally resulted in lower inference energy consumption.

**Impact of Network Depth:** Finally, we analyzed the impact of *TSkips* on inference energy across models of varying depth (Fig. 9(c)). Forward connections originated from the input, and backward connections originated from the final output, with $\Delta t = 16$. Our analysis revealed that increasing the number of layers led to a significant increase in inference energy.

This analysis of inference energy with varying *TSkips* parameters further emphasizes the complex interplay between $\Delta t$, *TSkips* position, and network depth. It highlights the importance of using methods like Kim et al. (2022) to find optimal network architectures and *TSkips* configurations. While Kim et al. (2022) does not explicitly optimize for minimal energy consumption, it prioritizes networks with diverse spiking activity by quantifying the dissimilarity between spiking patterns. As a result, the identified configurations often utilize *TSkips* with larger $\Delta t$ values, enabling our networks to efficiently capture long-term patterns while maintaining low energy consumption.

## B  Additional Results - DSEC-flow

### B.1 Inference Energy Analysis

As demonstrated in Section 4.3, *TSkips* reduce average endpoint error (AEE) without increasing model complexity or energy consumption for optical flow estimation on the DSEC-flow dataset. To further analyze this, we evaluated the inference energy of our fully spiking EV-FlowNet architectures augmented with *TSkips* using Eq. 6.

Table 5 demonstrates the effectiveness of incorporating *TSkips* into fully spiking EV-FlowNet architectures (Kosta & Roy, 2023). By efficiently utilizing temporal information, our approach achieves comparable

Table 5: Comparison of inference energy ($E_{total}$) using Eq. 6 for fully spiking DSEC-flow models with and without forward/backward TSkips across all scales.

| Model | #Params ($\times 10^6$) ($\mathcal{N}$) | Variant | #OPS$_{\text{SNN}}$ ($\times 10^9$) | Spike Rate ($\mathcal{M}$) | $E_{total}$ (mJ) |
|---|---|---|---|---|---|
| Base | 13.04 | Baseline | 25.9 | 45.83 | **23.3** |
| | | Forward *TSkips* | 32.7 | 57.8 | 29.4 |
| | | Backward *TSkips* | 30.7 | 54.32 | 27.6 |
| Mini | 3.41 | Baseline | 9.71 | 45.15 | 8.75 |
| | | Forward *TSkips* | 11.7 | 54.23 | **10.5** |
| | | Backward *TSkips* | 11.4 | 53.21 | **10.3** |
| Micro | 0.93 | Baseline | 5.81 | 61.9 | 5.25 |
| | | Forward *TSkips* | 6.14 | 65.4 | 5.52 |
| | | Backward *TSkips* | 5.86 | 62.4 | 5.27 |
| Nano | 0.27 | Baseline | 2.67 | 55.55 | 2.40 |
| | | Forward *TSkips* | 2.81 | 58.47 | 2.53 |
| | | Backward *TSkips* | 2.71 | 56.3 | 2.44 |
| Pico | 0.092 | Baseline | 2.11 | 74.12 | 1.90 |
| | | Forward *TSkips* | 2.29 | 80.6 | 2.07 |
| | | Backward *TSkips* | 2.25 | 78.9 | 2.02 |

AEE to baseline models 3× larger, while significantly reducing inference energy. This improvement stems from *TSkips* ability to enhance spike propagation with minimal added parameters. While adding *TSkips* increases the average spike rate, the resulting models achieve comparable AEE to larger baselines (Table 1) with substantially lower energy consumption. This highlights the potential of our method for accurate, robust, and efficient optical flow prediction.

## B.2   Inference Accuracy Analysis

As detailed in Sec. 4.5 the trained *TSkips* models were evaluated at inference after removing the temporal hidden state propagation through the *TSkips*. We provide qualitative results for these models in Table 6 and see that the DSEC models retain performance even without explicitly storing temporal states at inference. We attribute this to the learnable scaling factor ($\alpha$) within the *TSkips* as detailed in 3.3. This ($\alpha$) dynamically determines the weighted contribution of both the current timestep's information and the information from the delayed past timestep propagating through the skip connection. For our "no *TSkips*" inference analysis on DSEC, we used the learned $\alpha$ to control how much of the past hidden state's contribution was zeroed out in the skip connection, allowing the model to still leverage the learned weighting of the current timestep's information.

Table 6: Comparison of AEE (lower is better) on DSEC-flow across baseline architectures, models with forward (F)/ backward *TSkips* and models no *TSkips*.

| Arch | Baseline | F *TSkips* | F no *TSkips* | B *TSkips* | B no *TSkips* |
|---|---|---|---|---|---|
| Base | 1.35 | 1.12 | 1.13 | 1.13 | 1.13 |
| Micro | 1.65 | 1.44 | 1.44 | 1.46 | 1.47 |
| Mini | 1.80 | 1.57 | 1.58 | 1.56 | 1.56 |
| Nano | 2.17 | 1.86 | 1.86 | 1.77 | 1.77 |
| Pico | 2.57 | 2.19 | 2.20 | 2.28 | 2.30 |

### B.3 Convergence Analysis

As discussed in Section 3, incorporating *TSkips* can lead to easier training and faster convergence for both fully spiking (Kosta & Roy, 2023)and hybrid (Negi et al., 2024) EV-FlowNet architectures. To validate this, we analyzed the convergence behavior of our DSEC-flow models across different scales. Fig. 10 shows that both our largest (Base vs. Mini with *TSkips*) and smallest (Nano vs. Pico with *TSkips*) models exhibit faster convergence when incorporating *TSkips*, for both fully spiking and hybrid variants. This faster convergence can be attributed to *TSkips* ability to mitigate vanishing spikes and enhance gradient flow throughout the network, enabling quicker optimization and improved learning dynamics. Additionally, by facilitating the propagation of temporally relevant information, *TSkips* can help the network learn important temporal dependencies more efficiently, leading to faster convergence. This consistent trend across different model sizes highlights the benefits of *TSkips* in facilitating fast, efficient and effective training.

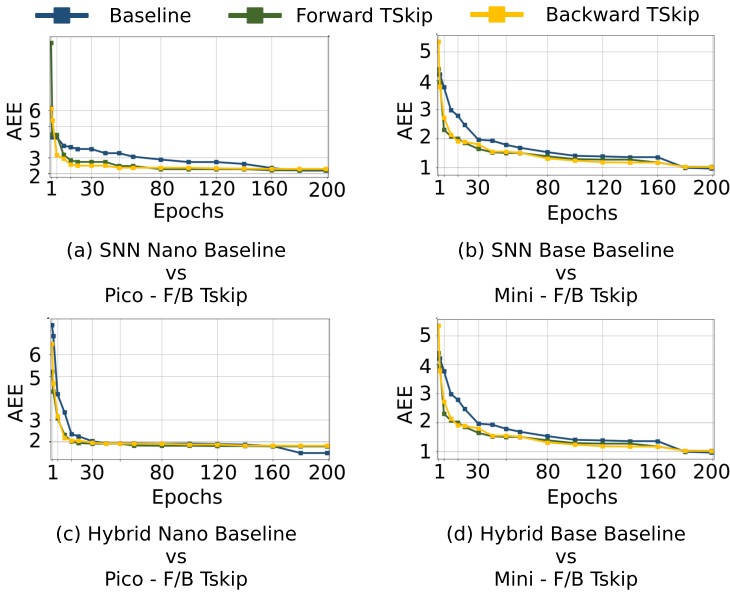

Figure 10: Convergence analysis of fully spiking (Kosta & Roy, 2023) and hybrid (Negi et al., 2024) EV-FlowNet architectures on the DSEC-flow dataset. The plots compare the convergence behavior of baseline models (Base and Nano) with their smaller forward (F) and backward (B) *TSkips* -augmented counterparts (Mini and Pico, respectively).

### B.4 Hyper parameters for DSEC-flow

Table 7 specifies the *TSkips* configurations used to replace existing skip connections in the fully spiking (Kosta & Roy, 2023) and hybrid (Negi et al., 2024) EV-FlowNet architectures. Here, "*TSkips* Position" indicates the targeted connection, with '1' representing the longest skip (between the first encoder and last decoder blocks) and '3' the shortest (between the last encoder and first decoder blocks), see Fig. 5.

## C Additional Results - DVS128 Gesture, SHD and SSC

### C.1 Inference Energy Analysis

As demonstrated in Section 4.4,*TSkips* improves classification accuracy for speech recognition compared to standard recurrent models (vRNNs and LSTMs). Additionally, we state that *TSkips* also reduce inference energy compared to these models. This energy efficiency is further confirmed by the estimations presented in Table 8 for both shallow (4-layer) and deep (8-layer) MLP (Rumelhart et al., 1986) networks on the SHD (Cramer et al., 2022) dataset. These estimations were calculated using Eq. 6, which considers the

Table 7: *TSkip* parameters used for fully spiking and hybrid DSEC-flow models.

| Architecture | SNN | | Hybrid | |
|---|---|---|---|---|
| | $\Delta t$ | *TSkip* Pos | $\Delta t$ | *TSkip* Pos |
| Base + Forward *TSkips* | 3 | 1 | 3 | 1 |
| Base + Backward *TSkips* | 4 | 1 | 5 | 1 |
| Mini + Forward *TSkips* | 3 | 1 | 4 | 2 |
| Mini + Backward *TSkips* | 3 | 2 | 3 | 1 |
| Micro + Forward *TSkips* | 4 | 2 | 3 | 2 |
| Micro + Backward *TSkips* | 3 | 3 | 3 | 3 |
| Nano + Forward *TSkips* | 3 | 1 | 3 | 2 |
| Nano + Backward *TSkips* | 4 | 2 | 3 | 2 |
| Pico + Forward *TSkips* | 4 | 1 | 3 | 1 |
| Pico + Backward *TSkips* | 3 | 2 | 4 | 2 |

energy consumed by both sparse accumulate (AC) operations in SNNs and dense multiply-and-accumulate (MAC) operations in ANNs.

For comparison, the vRNNs in Table 8 are our baseline SNNs with backward skip connections added to every layer with $\Delta t = 1$. The LSTMs are implemented as ANNs, with the $0.2 \times 10^6$ and $1.1 \times 10^6$ parameter models having 700 input channels and hidden sizes of 75 and 400, respectively.

The observed reduction in energy consumption can be attributed to several key factors. Firstly, *TSkips* introduce a limited number of additional synaptic connections ($\mathcal{C}$) compared to the densely connected nature of RNNs and LSTMs. Secondly, they promote sparse spiking activity by selectively propagating information across time steps, leading to lower energy consumption than the dense computations inherent in recurrent layers. Finally, the explicit temporal delays in *TSkips* enable the network to focus on the most relevant information from previous time steps, thereby reducing unnecessary computations and further improving energy efficiency.

Table 8: Inference energy ($E_{total}$) best-performing models on the SHD dataset. Energy is calculated using Eq. 6.

| Model | #Params $(\times 10^6)$ $(\mathcal{N})$ | #OPS $(\times 10^{12})$ | Spike Rate $(\times 10^4)$ $(\mathcal{M})$ | $E_{\textbf{total}}$ (mJ) |
|---|---|---|---|---|
| Baseline - 1 | 0.16 | 0.0083 | 3.16 | 7.52 |
| vRNN - 1 | 0.16 | 0.213 | 38.08 | 192.41 |
| LSTM - 1 | 0.2 | 0.016 | - | 74.63 |
| Forward *TSkips* -1 | 0.24 | 0.038 | 5.62 | 34.79 |
| Backward *TSkips* - 1 | 0.19 | 0.022 | 4.95 | 20.37 |
| Forward + Backward *TSkips* - 1 | 0.20 | 0.032 | 7.42 | 29.06 |
| Baseline - 2 | 1.04 | 0.815 | 7.63 | 734.26 |
| vRNN - 2 | 1.04 | 10.854 | 58.23 | 9768.88 |
| LSTM - 2 | 1.1 | 1.196 | - | 5505.28 |
| Forward *TSkips* - 2 | 1.12 | 1.204 | 8.21 | 1083.90 |
| Backward *TSkips* - 2 | 1.16 | 1.213 | 8.23 | 1091.87 |
| Forward + Backward *TSkips* - 2 | 1.28 | 1.761 | 9.13 | 1585.70 |

Table 9: Comparison of accuracy on SHD for two baseline models, forward (F)/ backward *TSkips* and no *TSkips*.

| Model | TSkips | no TSkips |
|---|---|---|
| Baseline - 1 | | 84.32 |
| FTSkips - 1 | 92.32 | 90.71 |
| BTSkips - 1 | 93.64 | 92.78 |
| F+B TSkips - 1 | 93.01 | 91.76 |
| Baseline - 2 | | 86.42 |
| FTSkips - 2 | 94.15 | 92.32 |
| BTSkips - 2 | 93.86 | 93.18 |
| F+B TSkips - 2 | 94.73 | 92.34 |

### C.2    Inference Accuracy Analysis

As detailed in Sec. 4.5 the trained *TSkips* models were evaluated at inference after removing the temporal hidden state propagation through the *TSkips*. We see that the SHD models without the *TSkips* have a slight decrease in inference accuracy. We attribute this slight decrease to less information being propagated to the later layers of the network through the *TSkips*. However, the models continue to perform better than the baselines, thus validating that *TSkips* helped to improve learning during training.

### C.3    Convergence Analysis

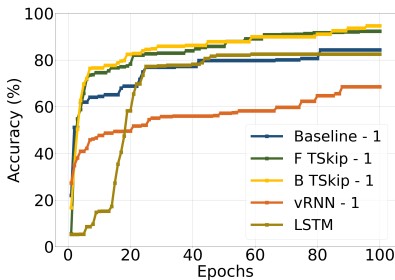

Figure 11: Test accuracy convergence on the SHD dataset, comparing the best-performing models with *TSkips* (from Table 3) to a vRNN and an LSTM with $0.2 \times 10^6$ parameters.

As discussed in Sections 3 and 4.4, incorporating *TSkips* can lead to easier training and faster convergence compared to recurrent models like vRNNs and LSTMs, as demonstrated in Fig. 11 for a 4-layer MLP network on the SHD dataset. This accelerated convergence can be attributed to several factors: improved gradient flow by mitigating the vanishing gradient problem, efficient information propagation by facilitating the access to temporally relevant information, and reduced computational complexity compared to the dense connectivity of recurrent networks. These factors, combined with the ability of *TSkips* to capture long-term temporal dependencies, contribute to the improved training efficiency observed in our experiments.

### C.4    TSkips in ANNs and Hybrid Models

As discussed in Section 4.4, incorporating *TSkips* into ANNs and hybrid ANN-SNN models can lead to improved classification accuracy. Table 10 presents these results, demonstrating the effectiveness of *TSkips* in enhancing both ANN and hybrid model performance, surpassing the performance of vRNNs and LSTMs, respectively.

This improvement can be attributed to the ability of *TSkips* to facilitate more effective temporal processing. By incorporating explicit temporal delays, these connections allow the network to access and integrate information from past time steps, which can be crucial for understanding complex temporal patterns. This enhanced temporal awareness enables the network to make more accurate predictions, leading to improved

Table 10: Classification accuracy of ANN and Hybrid ANN-SNN models on the SHD and SSC datasets. We compare the 4-layer MLP baseline (Baseline-1) with forward/backward *TSkips*.

| Method | #Params ($\times 10^6$) | Accuracy (%) | |
|---|---|---|---|
| | | ANN | Hybrid |
| **SHD** | | | |
| Baseline | 0.16 | 49.95 | **74.11** |
| vRNN | 0.16 | 60.51 | 67.45 |
| Forward *TSkips* | 0.24 | 69.08 | 84.54 |
| Backward *TSkips* | 0.19 | 73.45 | 86.02 |
| Forward + Backward *TSkips* | 0.20 | 73.25 | **85.74** |
| LSTM | 0.20 | 80.54 | - |
| **SSC** | | | |
| Baseline | 0.12 | 55.41 | **59.61** |
| vRNN | 0.12 | 68.29 | 71.43 |
| Forward *TSkips* | 0.42 | 71.19 | 73.48 |
| Backward *TSkips* | 0.24 | 70.23 | 72.87 |
| Forward + Backward *TSkips* | 0.54 | 71.09 | **74.96** |
| LSTM | 0.20 | 72.3 | - |

classification accuracy. Furthermore, *TSkips* can mitigate the vanishing spikes problem often encountered in deep networks, by providing more direct pathways for gradient flow during training. This improved gradient propagation can lead to more stable and efficient training, further contributing to the enhanced performance observed in ANNs and hybrid models.

### C.5 Comparison between TSkip Operators

To further analyze the impact of different *TSkips* connection types, we compared the performance of concatenation-based and addition-based *TSkips* on the 4-layer and 8-layer baselines (Table 3).

Table 11: Comparison of classification accuracy on SHD and SSC datasets with concatenation-based (C) and addition-based (A) forward/backward *TSkips* on a 4-layer (Baseline -1) and 8-layer (Baseline - 2) MLP network.

| Method | #Params ($\times 10^6$) | | Accuracy (%) | |
|---|---|---|---|---|
| | SHD | SSC | SHD | SSC |
| **Baseline - 1** | | | | |
| Baseline | 0.16 | 0.12 | 84.32 | 64.19 |
| Forward *TSkips* - C | 0.24 | 0.42 | 92.32 | 76.5 |
| Backward *TSkips* - C | 0.19 | 0.24 | 93.64 | 79.87 |
| Forward + Backward *TSkips* - C | 0.20 | 0.54 | 93.01 | 78.64 |
| Forward *TSkips* - A | 0.19 | 0.37 | 87.65 | 72.13 |
| Backward *TSkips* - A | 0.11 | 0.21 | 87.54 | 73.15 |
| Forward + Backward *TSkips* - A | 0.11 | 0.40 | 88.13 | 75.12 |
| **Baseline - 2** | | | | |
| Baseline | 1.04 | 1.04 | 86.42 | 67.54 |
| Forward *TSkips* - C | 1.12 | 1.08 | 94.15 | 78.98 |
| Backward *TSkips* - C | 1.16 | 1.14 | 93.86 | 79.65 |
| Forward + Backward *TSkips* - C | 1.28 | 1.42 | **94.73** | **80.23** |
| Forward *TSkips* - A | 0.99 | 1.05 | 86.39 | 72.15 |
| Backward *TSkips* - A | 0.97 | 1.01 | 87.32 | 75.53 |
| Forward + Backward *TSkips* - A | 0.98 | 1.14 | **89.45** | **77.12** |

Table 11 demonstrates the superior performance of concatenation-based *TSkips* over addition-based on the baseline models. Concatenation-based *TSkips* consistently outperformed their addition-based counterparts. This can be attributed to the increased representational capacity of concatenation, which expands the feature space by combining information from the skip connection and the destination layer. This richer representation allows the network to learn more complex temporal patterns, particularly crucial in deep SNNs where information is encoded in sparse spike trains and membrane potentials. Furthermore, concatenation preserves information from both sources, preventing potential information loss that can occur with addition.

### C.6 Hyper parameters for DVS128 Gesture, SHD and SSC

Table 13 presents the top-ranked network architectures and *TSkips* configurations identified by NASWOT-SAHD (Kim et al., 2022), with corresponding results shown in Tables 2 and 3.
The table specifies the following for each configuration:

1. Network architecture:
   (a) DVS128 Gesture: $2 \times 64 \times 64$ represents the input dimensions (polarity 2, $64 \times 64$ spike resolution) for all networks. Convolutional layers are represented concisely, for example, 3c80s1 denotes a layer with a $3 \times 3$ kernel, 80 output channels, and a stride of 1.
   (b) SHD and SSC: Specifies the number of input channels for each MLP layer, including the final fully connected layer. The channels represented as $n * 2$ indicate that concatenation-based *TSkips* were used, doubling the number of channels ($n$) in that layer. This representation is used to highlight that the results shown in Table 11, which uses addition-based *TSkips*, are based on the same underlying network architectures.

2. *TSkips* parameters: The temporal delay ($\Delta t$) and the origin and destination layers ($from \rightarrow to$) for forward (F), backward (B) and combined (F + B) *TSkips*.

3. Other hyper parameters:
   (a) Leak and threshold initialization values for adaptive LIF neurons: 0.6 and 15, respectively.
   (b) Dataset-specific dropout rates: 0.4 for SHD and 0.2 for SSC.

## D NASWOT-SAHD Validation

### D.1 Correlation on TSkips

To validate the effectiveness of NASWOT-SAHD (Kim et al., 2022) in identifying optimal network configurations, we analyzed the correlation between the accuracy and score of the identified networks, as shown in Fig. 12. The scores are based on the Sparsity Aware Hamming Distance (SAHD) (Kim et al., 2022), and the correlation is measured using Kendall's $\tau$. Our baseline models achieved a $\tau$ correlation of 0.63, consistent with the results reported in Kim et al. (2022). However, the forward and backward *TSkips* network searches

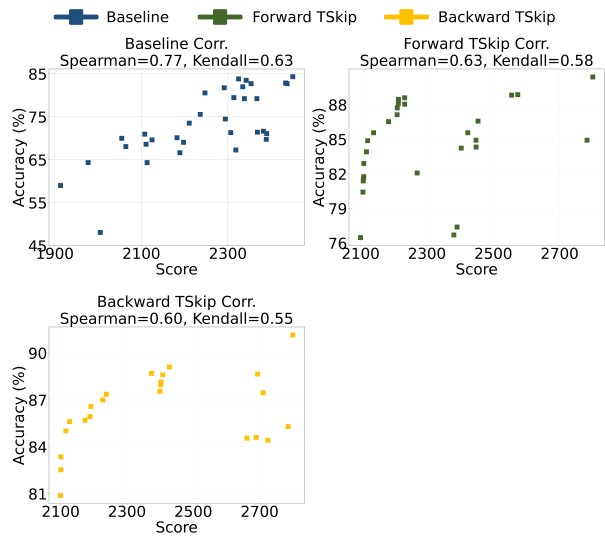

Figure 12: Correlation between accuracy and SAHD score for different network configurations (baseline, forward *TSkips*, and backward *TSkips*).

resulted in slightly lower cor-
relations of 0.58 and 0.55, respectively. All results reported are for the top-ranked networks identified by NASWOT-SAHD (Kim et al., 2022).

To ensure a focused and efficient search, we introduced several constraints:

1. Temporal Delay ($\Delta t$):

    (a) DSEC-flow ($T = 10$): $2 \leq \Delta t \leq 6$
    (b) DVS Gesture ($T = 30$): $5 \leq \Delta t \leq 14$
    (c) SHD and SSC ($T = 99$): $10 \leq \Delta t \leq 45$

2. *TSkips* Position: We constrained connections to not be within the same layer, ensuring that all connections originated from a different layer than the destination.

3. Model Size:

    (a) DSEC-Flow: No constraint was applied as we directly replaced existing skip connections in the fully spiking(Kosta & Roy, 2023) and hybrid (Negi et al., 2024) EV-FlowNet architecture with *TSkips*, maintaining the original model size.
    (b) DVS128 Gesture: $0.6 \times 10^6$ parameters
    (c) SHD and SSC (Baseline-1): $0.3 \times 10^6$ parameters
    (d) SHD and SSC (Baseline-2): $1.3 \times 10^6$ parameters

### D.2 Robustness across Random Seeds

To validate the robustness of our NAS-identified configurations and the consistent performance enhancement provided by *TSkips*, we conducted additional experiments on the DSEC-flow (Gehrig et al., 2021a) dataset. Table 12 presents the Average Endpoint Error (AEE) for both baseline models and their *TSkips*-augmented variants, averaged across three different random seeds. These results, reported as mean ($\mu$)$\pm$ standard deviation ($\sigma$), demonstrate the stability and reliable improvement offered by our approach.

Table 12: DSEC flow results on performing NAS search for *TSkips* with multiple random seeds. Presented below is the AEE (mean($\mu$) $\pm$ standard deviation($\sigma$)) (lower is better) of the models found across 3 random seeds.

| Model | $\Delta t$ | Position | **AEE** ($\mu \pm \sigma$) |
|---|---|---|---|
| Base - Baseline | - | - | $1.38 \pm 0.09$ |
| Base - FTSkip | 3,2,3 | 1,1,2 | $1.16 \pm 0.10$ |
| Base - BTSkip | 4,3,4 | 1,2,1 | $1.14 \pm 0.13$ |
| Mini - Baseline | - | - | $1.72 \pm 0.07$ |
| Mini - FTSkip | 3,2,2 | 1,1,2 | $1.49 \pm 0.06$ |
| Mini - BTSkip | 3,2,3 | 2,1,2 | $1.47 \pm 0.07$ |
| Micro - Baseline | - | - | $1.97 \pm 0.11$ |
| Micro - FTSkip | 4,3,3 | 2,1,2 | $1.66 \pm 0.10$ |
| Micro - BTSkip | 3,2,2 | 3,2,3 | $1.69 \pm 0.15$ |
| Nano - Baseline | - | - | $2.23 \pm 0.17$ |
| Nano - FTSkip | 3,3,2 | 1,1,2 | $1.85 \pm 0.07$ |
| Nano - BTSkip | 4,3,3 | 2,1,2 | $1.78 \pm 0.13$ |
| Pico - Baseline | - | - | $2.64 \pm 0.08$ |
| Pico - FTSkip | 4,2,3 | 1,2,1 | $2.32 \pm 0.13$ |
| Pico - BTSkip | 3,2,3 | 2,1,1 | $2.28 \pm 0.17$ |

Table 13: Optimal Parameters for Training MLP Networks with Temporal Skip Connections on SHD and SSC Datasets.

| Variant | Network | Delay ($\Delta t$) | TSkip Position (from → to) |
|---|---|---|---|
| **DVS128 Gesture** | | | |
| Baseline | 2×64×64-3c80s1-3c80s1-5c86s1-5c64s1-1c64s1-1c32s11 | - | - |
| F *TSkips* | 2×64×64-3c83s1-3c83s1-5c110s1-3c94s1-1c94s1-1c32s11 | 5 | 1 → 2 |
| B *TSkips* | 2×64×64-3c64s1-5c64s1-5c64s1-5c76s1-1c76s1-1c32s11 | 5 | 4 → 3 |
| F + B *TSkips* | 2×64×64-3c122s1-5c122s1-3c122s1-5c64s1-1c64s1+1c32s11 | F: 6 B: 8 | F: 1 → 3 B: 5 → 4 |
| **SHD** | | | |
| Baseline - 1 | 700-124-288-144-20 | - | - |
| F *TSkips* - 1 | 700-124*2-432-115-20 | 24 | 1 → 2 |
| B *TSkips* - 1 | 700*2-124-115-96-20 | 14 | 4 → 1 |
| F + B *TSkips* - 1 | 700*2-124-144-72*2-20 | F: 18 B: 14 | F: 2 → 4 B: 2 → 1 |
| Baseline - 2 | 700-512-288-224-192-896-288-20 | - | - |
| F *TSkips* - 2 | 700-532-288-524-192-448*2-288-20 | 17 | 2 → 6 |
| B *TSkips* - 2 | 700-321-558*2-334-292-452-328-20) | 12 | 3 → 5 |
| F + B *TSkips* - 2 | 700-321-228*2-334-492*2-462-448-20 | F: 10 B: 13 | F: 2 → 5 B: 7 → 3 |
| **SSC** | | | |
| Baseline - 1 | 700-124-148-130-35 | - | - |
| F *TSkips* - 1 | 700-324-298*2-145-35 | 15 | 1 → 3 |
| B *TSkips* - 1 | 700-184*2-179-230-35 | 12 | 3 → 2 |
| F + B *TSkips* - 1 | 700-364*2-257*2-195-35 | F: 14 B: 8 | F: 1 → 3 B: 4 → 2 |
| Baseline - 2 | 700-731-410-340-150-184-58-35 | - | - |
| F *TSkips* - 2 | 700-646-410-400-250-184*2-135-35 | 17 | 2 → 6 |
| B *TSkips* - 2 | 700-663-373*2-337-292-182-135-35 | 15 | 7 → 3 |
| F + B *TSkips* - 2 | 700-674-493*2-347*2-278-192-89-35 | F: 9 B: 11 | F: 1 → 2 B: 7 → 3 |

