# OpenReview forum: "TSkips: Efficiency Through Explicit Temporal Delay Connections in Spiking Neural Networks"
_TMLR — Accepted by TMLR_

### Review · Reviewer_edR7 · 2025-03-24

**Summary Of Contributions:**

The paper proposes a temporal skip connection augmentation to standard SNN architectures, with the idea that it enables learning of long-range temporal dependencies, resulting in improved performance on event-based datasets with smaller architectures.

**Audience:**

Yes

**Claims And Evidence:**

No

**Requested Changes:**

See weaknesses

**Strengths And Weaknesses:**

**Strengths**
-	The paper tackles an important problem of how SNNs can learn arbitrary range temporal dependencies that might be required for more accurate performance on event-based datasets, without needing heavy networks (like transformers) or adding too many complications in the network architecture.
-	Results on multiple datasets are promising for showing the value of the proposed approach.

**Weaknesses**
- I found the literature review and the positioning of the paper within the literature to be insufficient. My main concerns are:
  - Some important references are missing. I found at least a couple of them that I think are very relevant:
    - [Rethinking skip connections in Spiking Neural Networks with Time-To-First-Spike coding](https://doi.org/10.3389/fnins.2024.1346805)
    - [Efficient Processing of Spatio-Temporal Data Streams With Spiking Neural Networks](https://doi.org/10.3389/fnins.2020.00439)
  - I would want to see a discussion of these, and how the proposed method is different from these methods. I would also like to see a performance comparison.
  - A lot of SNN-related references in the paper seem to have the same subset of authors (Rathi, Roy, etc.). This is fine if these are being used as references in addition to other references, but for a lot of the claims in the Intro and throughout the paper, these are the only references. I would like to see a more diverse and wider range of references to substantiate the claims.
  - Some of the references do not seem to be relevant to the context in which they are being cited. For example, the paper Cao et al. 2015 is constantly cited to support the statement of vanishing gradients. However, I found no discussion of vanishing gradients in that paper.

- While the results are promising, I am not convinced that they support all the claims that the authors have made in the paper. For example, the following claims:
  - i) Offer finer control over spike timing
  - ii) Enhanced responsiveness to temporal patterns
  - iii) Even capturing long-range temporal dependencies
  These do not have any direct support in the results. The only support for these claims is the improvement in the downstream performance, but these claims are so specific that they will each require a more thorough analysis (similar to ablation) to validate. While it is okay to discuss these as possible reasons for improved performance, claiming that the proposed approach offers these benefits is too strong.

- There are several issues with experiments and results:
  - For the experiments on the DSEC-Flow dataset, E-RAFT gives a similar performance, but the paper says that it relies on complex mechanisms like gated recurrent units. This is too vague—what makes GRU more complex than Tskips + NAS? Likewise, the paper says that E-FlowFormer will “likely lead to higher inference energy than Tskips.” This is a conjecture that is not supported by any results. I would love to see an inference energy comparison of these two architectures. Additionally, what are mini, micro, and pico architectures mentioned in Table 1?
  - For DVS Gesture, SHD, and SSC datasets, the paper augmented the training sets with channel jitter and random noise. The paper should clarify if the baselines had the same augmentation too to ensure fair comparison.
  - The proposed method performs very similarly to DCLS-Delays, and the paper outlines several advantages of Tspikes over DCLS-Delays, including very few new trainable parameters. However, the total number of parameters of DCLS-Delays is the same as Tspikes and sometimes even less (Table 3) with greater performance. Given the difference in the fundamental components of DCLS-Delays and Tspikes, and the similar or higher performance of DCLS-Delays, I am not sure if Tspikes is a substitute for DCLS-Delays (as the paper indicates) or rather just a complementary approach.

- I would love to see a discussion of the intuition behind temporal skip connections and why they can enable long-range dependency learning, especially when combined with NAS, which can output skip connections between layers that are not too far from each other (indeed, some of the hyperparameters mentioned in the appendix seem to suggest so).

---

> ### Author Response · Authors · 2025-05-04
> **Official Comment by Authors**
>
> We thank the reviewer for their feedback and for helping improve our paper. We answer the questions from the reviewer below
>
> **W1.**
> - We have corrected the oversight in referencing works from a subset of authors and have included a broader range throughout the manuscript.
> - Additionally, Section 2 now includes a detailed explanation of how TSkips differs from the TTFS [1] method and [2]. Performance comparisons on the SHD dataset with TTFS are provided below:
>
> | Model          | SHD Accuracy (%) |
> |----------------|------------------|
> | Baseline - 1   | 84.32            |
> | FTSkips - 1    | 92.32            |
> | BTSkips - 1    | 93.64            |
> | F+B TSkips - 1 | 93.01            |
> | TTFS - 1       | 89.91            |
> | Baseline - 2   | 86.42            |
> | FTSkips - 2    | 94.15            |
> | BTSkips - 2    | 93.86            |
> | F+B TSkips - 2 | 94.73            |
> | TTFS - 2       | 90.12            |
>
> - We acknowledge our error in referencing  [3] for vanishing spikes and have replaced it with more relevant citations.
>
> #### [1] [Rethinking skip connections in Spiking Neural Networks with Time-To-First-Spike coding](https://doi.org/10.3389/fnins.2024.1346805)
> #### [2] [Efficient Processing of Spatio-Temporal Data Streams With Spiking Neural Networks](https://doi.org/10.3389/fnins.2020.00439)
>  #### [3] [# Spiking Deep Convolutional Neural Networks for Energy-Efficient Object Recognition](https://link.springer.com/article/10.1007/s11263-014-0788-3)
>
> **W2.** Section 4.5 in our updated manuscript addresses the validity of our claims, with supporting qualitative results and an analysis of inference accuracy with and without temporal information in the TSkips pathway provided in Appendices B.2 and C.2. This analysis confirms that trained TSkips models maintain superior performance even when the temporal information in the TSkips connection is removed during inference, underscoring their importance in enhanced learning. Specifically, Section 4.5 illustrates that the delayed spiking activity within TSkips during training (Fig. 8c, 8d) allows the model to learn temporal relationships, and their capacity to retain information from sparse, late-occurring events (Fig. 8a, 8b) indicates improved capture of long-term dependencies and a reduction in vanishing spikes.
>
> **W3.**
> 1.  We have clarified our comparisons to previous work on DSEC-flow:
>     -   **E-RAFT:** While E-RAFT uses GRUs, its higher computational complexity arises from extensive event pre-processing (voxel grids, CNN features, correlation volume), involving dense kernel operations for every event, unlike TSkips + NAS.
>     -   **E-FlowFormer:** Its higher inference energy is due to energy-intensive dense MAC operations in its Transformer architecture (self and cross-attention), required regardless of event sparsity. Its training also uses a large custom dataset alongside the DSEC-flow dataset. TSkips uses sparse AC operations, leveraging SNN sparsity, and avoids E-RAFT's heavy per-event pre-processing, suggesting lower inference energy despite potentially higher spike rates.
>     -   The different DSEC-flow architectures (Base, Mini, Micro, Nano, Pico) are scaled versions of the EV-FlowNet backbone with decreasing channel counts (64, 32, 16, 8, and 4, respectively), allowing us to assess TSkips' effectiveness across various model sizes.
> 2. Yes, for a fair comparison, the baseline models for DVS Gesture, SHD, and SSC were trained with the same channel jitter and random noise data augmentations as our TSkips models.
> 3.  We want to emphasize the distinct advantages of TSkips over methods like DCLS-Delays and clarify that we do not consider TSkips to be a direct substitute. DCLS-Delays necessitates modifying convolutional layers with dilated kernels, on the other hand TSkips can be integrated into any existing model architecture without requiring such layer-specific changes. Furthermore, TSkips achieves temporal learning through explicit skip connections without doubling the model's learnable parameters, offering a more parameter-efficient approach to enhance temporal processing in SNN designs. We have rephrased our writing in Section 4.4 to reflect this clarification.

---

> > ### Author Response · Authors · 2025-05-04
> > **Rebuttal Continued**
> >
> > **W4.** TSkips offer direct, efficient information flow across time, avoiding the signal degradation inherent in the sequential processing and backpropagation through time of standard recurrent networks. This direct access to past information helps mitigate vanishing/exploding gradients and enables the learning of selective temporal information. NASWOT-SAHD effectively finds good TSkips configurations and delay $(\Delta t)$ values because the proxy favors high linear separability in LIF activations across timesteps. This is vital for SNNs dealing with sparse event data, as high linear separability at initialization suggests neurons can learn distinct temporal features. We see that even with shorter inter-layer skips, NASWOT-SAHD identifies optimal $\Delta t$ values that maximize this linear separability along the temporal dimension.

---

### Review · Reviewer_caaK · 2025-04-01

**Summary Of Contributions:**

This paper introduces TSkips, a mechanism that enhances Spiking Neural Networks by incorporating explicit temporal delays in skip connections. The authors propose both forward and backward temporal skip connections that enable direct transmission of spike information between non-adjacent layers and different time steps. The exact skip connections are found using a training-free Neural Architecture Search.
The effectiveness of TSkips is evaluated on four event-based datasets: DSEC-flow for optical flow estimation, DVS128 Gesture for hand gesture recognition, and Spiking Heidelberg Digits and Spiking Speech Commands for speech recognition. Results show improvements across all datasets.

**Audience:**

Yes

**Claims And Evidence:**

No

**Requested Changes:**

Following up on the weaknesses I have mentioned in the previous questions, I would like the following points to be addressed to secure my recommendation for acceptance:

- Can the authors tune the learning rates in their different experiments (for all the results they produce)?
- Could the authors argue why applying NAS to different initializations would not yield benefits as significant as the ones they found with their method? One experiment to clarify that on, e.g., DSEC-flow would be welcomed.
- How can the TSkips be implemented on "physical" hardware? What are the related extra memory / energy costs?

The first two points are to the best of my understanding critical to ensure the robustness of the experiments, and the last one is an important consideration to clarify.

Less critical but still important is to include a reference to [1], which studied skip connections through time in rate based recurrent networks (with motivations similar to the ones here). Potentially of interest for my last point above: they removed the skip connection through time after training. Would the same benefits would hold for SNNs? If so, such a result would strengthen the paper: there is no need for extra mechanisms at inference time and the benefits would solely be attributed to learning (and not the addition of new pathways, as it may partially be the case in the current version).

[1] Training biologically plausible recurrent neural networks on cognitive tasks with long-term dependencies, Soo et al.

**Strengths And Weaknesses:**

### Strengths

- The paper is well-written: the motivation is clearly stated, the proposed approach is easy to understand, and all important details are clearly explained.
- The paper includes thorough analyses of how various parameters (temporal delay values, skip connection positions, network depth) affect model performance.
- The authors test their approach across multiple datasets and tasks (optical flow, gesture recognition, speech recognition), demonstrating the versatility and effectiveness of TSkips.

### Weaknesses

Overall, I do have some concerns about the robustness of the empirical results as well as some about "physical" implementations of the proposed mechanism. I provide an overview below and more details in the next section.
- As far as I can tell the learning rates were not tuned. While this seems to be the case both for the proposed approach and the baselines, I am afraid that this may increase the performance difference between the two.
- It is not clear to me how much of the performance gain is due to finding "easy to learn" initializations thanks to the training-free NAS. One concern would be that TSkips adds some variability to the architecture, which increases the chance that a good initialization exists. It could be that applying the NAS on different seeds for the baselines already boosts performance. Maybe even more when adding more variability, e.g. magnitude of the weights / thresholds... To the best of my understanding, this possibility is partially ruled out by the MLP ablations (as there is not NAS there (?)), but not really in the other experiments.
- There is no discussion of how physical networks (i.e. neuromorphic hardware / biological networks) could implement these mechanisms. To the best of my knowledge, these are one (if not the) main interests of spiking networks and the proposed mechanism requires some storing mechanisms that at best requires extra memory.

---

> ### Author Response · Authors · 2025-05-04
> **Official Comment by Authors**
>
> We thank the reviewer for their feedback and for suggesting experiments and clarifications that can improve our results and paper. We answer the questions from the reviewer below
>
> **W1./Q1.** We tuned the learning rates for all reported models. As detailed in Section 4.1, our training procedure involved both a multi-step learning rate scheduler and a Cosine Annealing Learning Rate Scheduler. Before implementing these schedulers, we performed a manual tuning phase across all datasets to identify a suitable initial learning rate.
>
> **W2./Q2.** The NASWOT-SAHD proxy favors good initializations with high linear separability, and the TSkips architecture modifications changes the linear separability of the basleines. Howveer, the benefits of TSkips go beyond favorable initializations. The temporal connections modify information flow and gradient propagation during training (Section 3.3), enabling the network to learn richer spatio-temporal features and capture longterm dependencies more effectively than baseline architectures, regardless of their initializations (we provide an analysis of this in Section 4.5.). To further support this, we evaluated NAS-identified TSkips configurations on DSEC-flow across two additional random seeds, and the consistent performance improvements (mean ± standard deviation) over baseline models, as detailed in the table below, confirm the performance gains with TSkips.
>
> | Model             | $\Delta t$ | Position | AEE ($\mu \pm \sigma$) (lower is better) |
> |-------------------|-------------|----------|------------------------------------------|
> | Base - Baseline   | -           | -        | 1.38 $\pm$ 0.09                            |
> | Base - FTSkip     | 3,2,3       | 1,1,2    | 1.16 $\pm$ 0.10                            |
> | Base - BTSkip     | 4,3,4       | 1,2,1    | 1.14 $\pm$ 0.13                            |
> | Mini - Baseline   | -           | -        | 1.72 $\pm$ 0.07                            |
> | Mini - FTSkip     | 3,2,2       | 1,1,2    | 1.49 $\pm$ 0.06                            |
> | Mini - BTSkip     | 3,2,3       | 2,1,2    | 1.47 $\pm$ 0.07                            |
> | Micro - Baseline  | -           | -        | 1.97 $\pm$ 0.11                            |
> | Micro - FTSkip    | 4,3,3       | 2,1,2    | 1.66 $\pm$ 0.10                            |
> | Micro - BTSkip    | 3,2,2       | 3,2,3    | 1.69 $\pm$ 0.15                            |
> | Nano - Baseline   | -           | -        | 2.23 $\pm$ 0.17                            |
> | Nano - FTSkip     | 3,3,2       | 1,1,2    | 1.85 $\pm$ 0.07                            |
> | Nano - BTSkip     | 4,3,3       | 2,1,2    | 1.78 $\pm$ 0.13                            |
> | Pico - Baseline   | -           | -        | 2.64 $\pm$ 0.08                            |
> | Pico - FTSkip     | 4,2,3       | 1,2,1    | 2.32 $\pm$ 0.13                            |
> | Pico - BTSkip     | 3,2,3       | 2,1,1    | 2.28 $\pm$ 0.17                            |
>
> **W3./Q3.** TSkips can be implemented on standard hardware by storing and retrieving past layer states, incurring memory and energy costs for simulation. On neuromorphic hardware, TSkips can be realized through direct routing with delays, where information from an earlier layer is directly routed to a later layer with a controlled temporal delay. This can be achieved using physical delay lines or digital buffers within the chip's architecture, potentially offering energy efficiency by bypassing computations in intermediate layers. Importantly, our findings in Section 4.5 show that trained TSkips models can perform well even without explicitly using the temporal skip at inference, reducing hardware memory and energy requirements. We provide a discussion of this in Section 5 in our updated manuscript.

---

> > ### Author Response · Authors · 2025-05-04
> > **Rebuttal Continued**
> >
> > **Q4.**  To see if the benefits of training with TSkips remain when temporal state storage is removed at inference (as shown in [1] for rate-based networks), we tested our trained DSEC-flow and SHD MLP models. For the "no TSkips" models, we passed a null tensor through the skip connection.
> >
> > #### DSEC-flow (Lower AEE is better)
> > | Arch  | Baseline | FTSkips | F no TSkips | BTSkips | B no TSkips |
> > |-------|----------|---------|-------------|---------|-------------|
> > | Base  | 1.35     | 1.12    | 1.13        | 1.13    | 1.13        |
> > | Micro | 1.65     | 1.44    | 1.44        | 1.46    | 1.47        |
> > | Mini  | 1.80     | 1.57    | 1.58        | 1.56    | 1.56        |
> > | Nano  | 2.17     | 1.86    | 1.86        | 1.77    | 1.77        |
> > | Pico  | 2.57     | 2.19    | 2.20        | 2.28    | 2.30        |
> >
> > #### SHD (Accuracy (%) )
> > | Model          | TSkips | no TSkips |
> > |----------------|--------|-----------|
> > | Baseline - 1   | 84.32  |           |
> > | FTSkips - 1    | 92.32  | 90.71     |
> > | BTSkips - 1    | 93.64  | 92.78     |
> > | F+B TSkips - 1 | 93.01  | 91.76     |
> > | Baseline - 2   | 86.42  |           |
> > | FTSkips - 2    | 94.15  | 92.32     |
> > | BTSkips - 2    | 93.86  | 93.18     |
> > | F+B TSkips - 2 | 94.73  | 92.34     |
> >
> > - The DSEC models maintain performance well without explicit temporal state storage at inference. We attribute this to the learnable scaling factor  $(\alpha)$  in the TSkips dynamically weights the current and past timestep information. In our "no TSkips" DSEC analysis, we used the learned  $\alpha$  to effectively zero out the past hidden state's contribution, allowing the model to still leverage the learned weighting of the current timestep's information.
> > - Conversely, the SHD baseline has no skip connections. Therefore, in the "no TSkips" SHD analysis, we nulled the information that would have passed through the temporal skips, leading to a slight decrease in inference accuracy.
> >
> > [1] [Training biologically plausible recurrent neural networks on cognitive tasks with long-term dependencies, Soo et al.](https://proceedings.neurips.cc/paper_files/paper/2023/file/65ccdfe02045fa0b823c5fa7ffd56b66-Supplemental-Conference.pdf)

---

### Review · Reviewer_BsKM · 2025-04-23

**Summary Of Contributions:**

The paper introduces TSkips, a novel mechanism for Spiking Neural Networks (SNNs), which augments the network architecture with temporal skip connections that incorporate explicit temporal delays. TSkips are designed to capture long-term spatio-temporal dependencies by enabling direct transmission of spike information across non-adjacent layers. The authors employ a training-free Neural Architecture Search (NAS) to identify optimal configurations for TSkips, including temporal delays and connection placements, and demonstrate their effectiveness on four event-based datasets.

**Audience:**

Yes

**Claims And Evidence:**

No

**Requested Changes:**

1. A detailed analysis should be included to explain how TSkip connections help capture better spatio-temporal features. It would be beneficial to provide insights on which types of connections work best for capturing long-term temporal dependencies.

2. The experiments should address the scalability of TSkips, especially for deep networks and long temporal sequences. The paper should discuss how TSkips can be effectively used in larger, more complex networks.

3. To demonstrate the full potential of TSkips, the authors should consider testing the method on more challenging datasets. This would provide stronger evidence of the method’s robustness and versatility.

**Strengths And Weaknesses:**

## Strengths
1. The paper effectively tackles a crucial aspect of SNNs: the temporal domain. The exploration of temporal dependencies through TSkips is a key strength, as it directly enhances the model’s ability to capture spatio-temporal patterns, which is vital for tasks involving event-based data.

2. The application of NAS to explore the vast search space of TSkip.

3. The experimental results demonstrate clear performance improvements in accuracy and energy efficiency across different datasets, including optical flow, gesture recognition, and speech tasks.

## Weaknesses
1. While TSkips introduce a novel approach, the search space for optimal configurations (skip positions and time delays) is vast and difficult to navigate. The reliance on NAS introduces a dependency on the effectiveness of the search algorithm. This could undermine the novelty of TSkips since its performance is heavily influenced by the NAS method.

2. The method may face challenges when applied to deep SNN architectures with long temporal sequences. The increasing complexity of the skip connection search space with network depth and sequence length can be computationally expensive and difficult to manage.

3. The paper does not provide a detailed analysis on how TSkips contribute to capturing “spatio-temporal patterns” as claimed. Specifically, there is no clear discussion on which connections capture better features and how TSkips enhance this process. A more thorough examination of this aspect would strengthen the claims.

4. The experiments involving parameter reduction (<10M) do not clearly justify the significance of scaling down models for TSkips. The real challenge lies in scaling up models with larger parameters (>50M), which would provide a more rigorous test of TSkip's effectiveness.

5. While the authors conduct experiments across multiple tasks, the difficulty of these tasks is relatively low. For instance, the DVS-Gesture dataset is considered simple, and testing on more challenging datasets such as DVS-Cifar100 or Tiny-N-Imagenet would provide a more robust validation of TSkip's capability.

6. There are instances of subjective claims without sufficient evidence. For example, in the introduction, the authors state: “We posit that these learnable synaptic delays only capture spatio-temporal patterns over a short window of time” without providing empirical or theoretical evidence to support this. These types of statements should be better substantiated to enhance credibility.

---

> ### Author Response · Authors · 2025-05-04
> **Official Comment by Authors**
>
> We thank the reviewer for their feedback and for helping improve our paper. We answer the questions from the reviewer below
>
> **Requested Changes**
> **Q1.** We refer the reviewer to Section 4.5 in our updated manuscript where we address the validity of our claims better. To summarize:
> (i) During training, TSkips enables the model to learn temporal relationships, as evidenced by the average spike representations in Fig. 8a and 8b. Trained TSkips models, both with and without past information, more closely follow the input event spike distribution compared to the baseline. In contrast, the baseline model only emits a spike after significant accumulation. This demonstrates that the time-shifted information within TSkips is crucial for effective learning via BPTT.
>
> (ii) The ability of TSkips models to better capture long-term dependencies is supported by observations in Fig. 8a and 8b. These figures show that both TSkips variants retain information from sparse, later input spikes, leading to a more accurate realization of these spikes in the tail end of the distribution. This suggests that the temporal shift $(\Delta t)$ introduced by TSkips helps overcome the vanishing spike problem more effectively than baseline models.
>
> **Q2.** We addressed the challenge of varying temporal sequence lengths by conducting experiments across datasets with different sequence lengths (T) (ranging from 99, 30 and 10 as detailed in our experimental setup). Furthermore, our architecture search explored a diverse range of models, including ResNet18, MLPs with varying depths, and an encoder-decoder architecture. To manage the increasing complexity of the search space in deeper networks and longer sequences, the exploration process is adapted by implementing constraints to control the similarity/dissimilarity between searched architectures and TSkips configurations. To validate that the NAS proxy finds near optimal solutions, we provide results on DSEC-flow for 2 additional random seeds with the TSkips configurations.
>
> | Model             | $\Delta t$ | Position | AEE ($\mu \pm \sigma$) (lower is better) |
> |-------------------|-------------|----------|------------------------------------------|
> | Base - Baseline   | -           | -        | 1.38 $\pm$ 0.09                            |
> | Base - FTSkip     | 3,2,3       | 1,1,2    | 1.16 $\pm$ 0.10                            |
> | Base - BTSkip     | 4,3,4       | 1,2,1    | 1.14 $\pm$ 0.13                            |
> | Mini - Baseline   | -           | -        | 1.72 $\pm$ 0.07                            |
> | Mini - FTSkip     | 3,2,2       | 1,1,2    | 1.49 $\pm$ 0.06                            |
> | Mini - BTSkip     | 3,2,3       | 2,1,2    | 1.47 $\pm$ 0.07                            |
> | Micro - Baseline  | -           | -        | 1.97 $\pm$ 0.11                            |
> | Micro - FTSkip    | 4,3,3       | 2,1,2    | 1.66 $\pm$ 0.10                            |
> | Micro - BTSkip    | 3,2,2       | 3,2,3    | 1.69 $\pm$ 0.15                            |
> | Nano - Baseline   | -           | -        | 2.23 $\pm$ 0.17                            |
> | Nano - FTSkip     | 3,3,2       | 1,1,2    | 1.85 $\pm$ 0.07                            |
> | Nano - BTSkip     | 4,3,3       | 2,1,2    | 1.78 $\pm$ 0.13                            |
> | Pico - Baseline   | -           | -        | 2.64 $\pm$ 0.08                            |
> | Pico - FTSkip     | 4,2,3       | 1,2,1    | 2.32 $\pm$ 0.13                            |
> | Pico - BTSkip     | 3,2,3       | 2,1,1    | 2.28 $\pm$ 0.17                            |

---

> > ### Author Response · Authors · 2025-05-04
> > **Rebuttal Continued**
> >
> > **Q3.** We have now evaluated TSkips on CIFAR10-DVS dataset using spiking ResNet18 (11.7M) and VGG11 (132.8M) models with the NDA [1] data augmentation technique. The results, presented below, demonstrate that TSkips continues to improves accuracy over the baseline models:
> >
> > | Model                     | Accuracy (%) |
> > |---------------------------|--------------|
> > | ResNet18 - NDA [1]        | 78.00        |
> > | ResNet18 NDA - FTSkips    | 81.08        |
> > | ResNet18 NDA - BTSkips    | 82.93        |
> > | VGG11 - NDA [1]           | 81.70        |
> > | VGG11 NDA - FTSkips       | 82.56        |
> > | VGG11 NDA - BTSkips       | 83.01        |
> >
> > - These results demonstrate TSkips' ability to enhance performance in deeper networks on a more complex visual task. However, we didn't see the same convergence benefits as on other datasets, likely because CIFAR10-DVS's temporal information is artificially generated. We believe these findings offer valuable insight into the use cases of TSkips' and its effectiveness. TSkips provides benefits in challenging vision/audio tasks, that have rich temporal information, as compared to the static images in CIFAR10-DVS that have been made dynamic.
> > - While CIFAR10-DVS is more complex than DVS-Gesture, we would like to clarify that DSEC-flow presents a more significant challenge due to its real-world driving sequences that inherently contain complex artifacts such as varying lighting, significant occlusions, and high resolution (640x480). Accurately estimating optical flow in such conditions demands the capture of intricate spatio-temporal dynamics present in event data, making it a more demanding task.
> > - A DVS version of Tiny Imagenet is not available and since static images do not have any temporal information we did not evaluate TSkips on it.
> >
> > [1] [Li, Yuhang, et al. "Neuromorphic data augmentation for training spiking neural networks." _European Conference on Computer Vision_. Cham: Springer Nature Switzerland, 2022.](https://link.springer.com/chapter/10.1007/978-3-031-20071-7_37)
> >
> > **Weaknesses**
> > **W1.** Like any architecture search  that relies on the proxy to find good initializations, TSkips relies on the training-free NAS proxy to find optimal TSkips configurations ($\Delta t$ and positions). However, the NAS proxy alone does not guarantee performance gains with TSkips. To demonstrate that TSkips' provides consistent improvement, we ask the reviewer to reference the two tables above.
> >
> > **W6.** We have removed the claim that "learnable synaptic delays only capture spatio-temporal patterns over a short window of time" from our updated manuscript as we cannot substantiate it sufficiently.

---

### Review · Reviewer_HneG · 2025-05-24

**Summary Of Contributions:**

This paper introduces TSkips, a novel architectural mechanism that augments Spiking Neural Networks (SNNs) and hybrid ANN-SNN models with forward and backward skip connections that include explicit temporal delays. The goal is to enhance the ability of SNNs to model long-term spatio-temporal dependencies in event-based data while improving computational efficiency.

**Audience:**

Yes

**Claims And Evidence:**

No

**Requested Changes:**

Stability Against Perturbations and Hardware Noise: To establish the robustness of TSkips in practical scenarios, we recommend the authors include a stability analysis under the following conditions:
1. Input perturbations (e.g., noise in event timing or channel jitter),
2. Internal perturbations (e.g., quantization noise, delayed or dropped spikes),

**Strengths And Weaknesses:**

Practical Advantages and Hardware Considerations:
1. TSkips introduce minimal overhead, with very few additional parameters.
2. Discusses potential for neuromorphic implementation using delay lines or buffers.
3. Highlights that TSkips can be disabled at inference time to save memory/energy with minimal performance drop.

---

> ### Author Response · Authors · 2025-07-02
>
> We thank the reviewer for their feedback and apologize for the delayed response.
>
> We appreciate the feedback regarding the stability analysis of TSkips against perturbations and hardware noise. We agree this is an important area for practical scenarios and neuromorphic implementations. This specific analysis leans towards adversarial robustness of TSkips which falls a bit outside the scope of the paper's focus and the minor revisions requested. This is an interesting avenue for future work.

---

### Decision · Action_Editor_46f4 · 2025-06-03

**Recommendation:** Accept with minor revision

**Comment:**

This paper presents an interesting architecture that enhances Spiking Neural Networks (SNNs) and hybrid ANN-SNN models. Following the discussion, two reviewers provided positive feedback (Accept, Leaning accept). Although some concerns were raised (e.g., scalability for deep networks, the need for more comparisons with SOTA methods), all reviewers acknowledged the novelty and contributions of the proposed method. The method is expected to attract significant interest from TMLR's readership, and thus I recommend “Accept with minor revision”. For the final version, the authors are required to carefully revise the manuscript and incorporate additional analysis, especially on the scalability of the architecture.

**Audience:**

Yes

**Claims And Evidence:**

Yes